# Floquet fractional Chern insulators and competing phases in twisted bilayer graphene

**Peng-Sheng Hu°, Yi-Han Zhou° and Zhao Liu⋆**

Zhejiang Institute of Modern Physics, Zhejiang University, Hangzhou 310058, China

⋆ zhaol@zju.edu.cn

## Abstract

We study the many-body physics in twisted bilayer graphene coupled to periodic driving of a circularly polarized light when electron-electron interactions are taken into account. In the limit of high driving frequency $\Omega$, we use Floquet theory to formulate the system by an effective static Hamiltonian truncated to the order of $\Omega^{-2}$, which consists of a single-electron part and the screened Coulomb interaction. We numerically simulate this effective Hamiltonian by extensive exact diagonalization in the parameter space spanned by the twist angle and the driving strength. Remarkably, in a wide region of the parameter space, we identify Floquet fractional Chern insulator states in the partially filled Floquet valence bands. We characterize these topologically ordered states by ground-state degeneracy, spectral flow, and entanglement spectrum. In regions of the parameter space where fractional Chern insulator states are absent, we find topologically trivial charge density waves and band-dispersion-induced Fermi liquids which strongly compete with fractional Chern insulator states.



## Contents

° These authors contributed equally to the development of this work.



# 1   Introduction

Van-der-Waals heterostructures with moiré patterns [1,2] have attracted tremendous attention over the last few years. When two atomic layers are stacked with each other, the mismatch between the two crystals due to different lattice constants and/or a twist angle generates a large-scale superlattice and affects the interlayer coupling. These moiré systems, the band structures of which are highly controllable, are promising hosts of numerous exotic phenomena. A representative example is the twisted bilayer graphene (TBG), consisting of two sheets of monolayer graphene with a twist angle in between [3]. One of the most striking features of TBG is that, at certain special twist angles (called magic angles), the low-energy bands near the charge neutrally point (CNP) can be tuned to be nearly flat, thus providing an ideal platform to investigate correlated physics. Indeed, unconventional superconductivity and ferromagnetism have been discovered in TBG [4–9]. Another salient progress in this direction is the observation of the quantum anomalous Hall effect at zero magnetic field (also called Chern insulators) in TBG aligned with a hexagonal boron nitride (hBN) [10]. In this case, the hBN gaps out the protected Dirac band touching of TBG, such that the flat bands around the CNP are isolated and acquire nonzero Chern numbers [6, 10–14]. Motivated by this topological band structure, a lot of theoretical efforts have been made to demonstrate the possibility of realizing the zero-field fractional Chern insulators (FCIs) [15–17] in TBG-hBN when the flat bands near the CNP are partially occupied by interacting electrons [18–21]. Excitingly, evidence of FCIs in TBG-hBN has been reported in a recent experiment at weak magnetic fields [22].

External driving fields of light provide an alternative avenue to obtain topological band structures even if the original bands in the absence of driving are topologically trivial [23–30]. Unlike nondriven systems, systems periodically driven by light do not have well-defined static band structures. However, the effect of light enters an effective static Hamiltonian that captures the dynamics of the system on time scales much longer than the driving period [24,31–33]. This effective static Hamiltonian, which describes photo-dressed band structures, has been extensively used to predict the out-of-equilibrium topological properties of various periodically driven systems [23,24,27,28,34]. In particular, accompanying the rapid development of moiré materials, exploring the effects of light on moiré band structures has become an intriguing direction recently [35–46]. Just like in monolayer graphene [23,24,47], a circularly polarized light can open a band gap at the Dirac points of TBG and give rise to Floquet topological flat bands [36–38].

While exciting progress has been made on Floquet moiré materials, the interactions between electrons are not yet taken into account in previous works. Hence whether correlated topological phases can be induced by light driving in these systems remains a crucial open question. In the context of photon-dressed topological band structure, a natural candidate of correlated topological states is the Floquet FCI which was first proposed in monolayer graphene [48, 49]. Floquet FCIs can be formed when interacting particles partially occupy a topological flat band of the effective static Hamiltonian of the driven system.

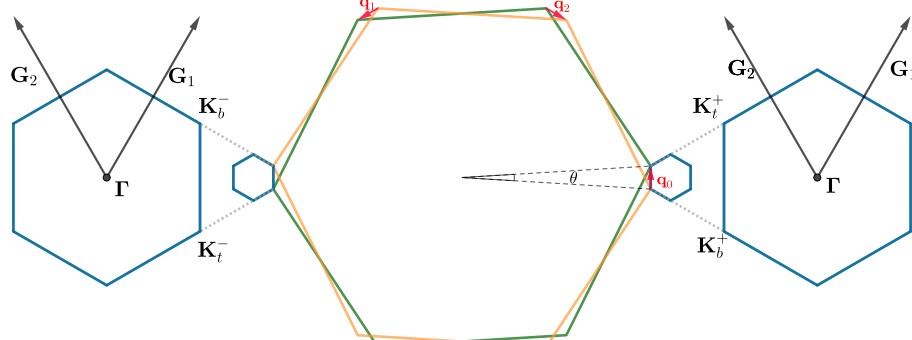

Figure 1: Moiré Brillouin zone of TBG. The two large hexagons are original first Brillouin zones of top (green) and bottom (orange) graphene monolayers. The two small hexagons represent the MBZs resulting from twist, corresponding to the two valleys of TBG. The $\mathbf{K}_\xi^{t,b}$ points, the vectors $\mathbf{q}_{0,1,2}$ in Eq. (1) and the MBZ reciprocal lattice vectors $\mathbf{G}_1$ and $\mathbf{G}_2$ are given.

In this work, we investigate the possibility of stabilizing Floquet FCIs in TBG driven by a circularly polarized laser light. We focus on the limit of high driving frequency $\Omega$ and derive the effective static Hamiltonian truncated to the order of $\Omega^{-2}$. Unlike Ref. [49], we find this effective static Hamiltonian does not contain new effective interactions that could diminish FCIs. This feature is due to the special form of the TBG Hamiltonian. Then we use exact diagonalization to search for the evidence of Floquet FCIs in the parameter space spanned by the twist angle and the driving strength. To be concrete, we assume that the Floquet valence bands in the two TBG valleys are occupied at total filling $\nu = 1/3$ by electrons interacting via the screened Coulomb potential. In this case, we find a wide region in which valley-polarized FCIs are promising to exist. We also observe charge density waves (CDWs) and Fermi liquid (FL) states in the neighboring regions of FCIs.

## 2 Model

We consider the low-energy dynamics of TBG at small twist angles ($\theta \sim 1°$) by following Bistritzer and MacDonald's continuum model [50]. After twisting, the original first Brillouin zone of monolayer graphene is folded into moiré Brillouin zones (MBZs). The low-energy states of electrons reside in the MBZs near the Dirac points $\mathbf{K}_+$ and $\mathbf{K}_-$ of monolayer graphene, called as two valleys of TBG which we denote by $\xi = \pm$ (Fig. 1). In valley $\xi$, the $\mathbf{K}_\xi$ points of the top and bottom graphene layers are located at $\mathbf{K}_\xi^t = R_{\theta/2}\mathbf{K}_\xi$ and $\mathbf{K}_\xi^b = R_{-\theta/2}\mathbf{K}_\xi$, respectively, with $R_\theta$ a counter-clockwise rotation by angle $\theta$ around the $z$-axis. $\mathbf{G}_1$ and $\mathbf{G}_2$ are the primitive reciprocal lattice vectors of the MBZ (Fig. 1), with $\mathbf{a}_1$ and $\mathbf{a}_2$ the corresponding real-space primitive lattice vectors. We neglect the intervalley scattering and spin-orbit coupling which are very weak at small twist angles.

## 2.1 Static system

In the absence of driving, the single-electron Hamiltonian of TBG takes the form of [18, 51]

$$H_{\text{kin}} = \sum_{\xi=\pm} \sum_{\mathbf{k}} \left( \psi_{t,\xi}^{\dagger}(\mathbf{k}) h_{-\theta/2}^{\xi}\left(\mathbf{k} - \mathbf{K}_{\xi}^{t}\right) \psi_{t,\xi}(\mathbf{k}) + \psi_{b,\xi}^{\dagger}(\mathbf{k}) h_{\theta/2}^{\xi}\left(\mathbf{k} - \mathbf{K}_{\xi}^{b}\right) \psi_{b,\xi}(\mathbf{k}) \right)$$

$$+ \sum_{\xi=\pm} \sum_{\mathbf{k}} \sum_{j=0}^{2} \left( \psi_{t,\xi}^{\dagger}\left(\mathbf{k} - \xi\mathbf{q}_0 + \xi\mathbf{q}_j\right) T_j^{\xi} \psi_{b,\xi}(\mathbf{k}) + \text{H.c.} \right), \tag{1}$$

where $\psi_{t,\xi}(\mathbf{k}) = \begin{pmatrix} \psi_{tA,\xi}(\mathbf{k}) \\ \psi_{tB,\xi}(\mathbf{k}) \end{pmatrix}$ and $\psi_{b,\xi}(\mathbf{k}) = \begin{pmatrix} \psi_{bA,\xi}(\mathbf{k}) \\ \psi_{bB,\xi}(\mathbf{k}) \end{pmatrix}$ are spinors of annihilation operators in valley $\xi$ for electrons in top ($t$) and bottom ($b$) graphene layers, respectively, and $A$ and $B$ correspond to the two sublattices in monolayer graphene. The first line in Eq. (1) includes the Dirac Hamiltonians of top and bottom graphene layers, for which $h_{\theta}^{\xi}(\mathbf{k}) = h^{\xi}(R_{\theta}\mathbf{k})$ with $h^{\xi}(\mathbf{k}) = -\hbar v_F(\xi k_x \sigma_x + k_y \sigma_y)$. Here $\sigma_{x,y,z}$ are the Pauli matrices acting on the sublattice degree of freedom. We set $\hbar v_F = \sqrt{3}at_0/2$, where $t_0 = 2.62$ eV is the nearest-neighbor hopping amplitude and $a = 0.246$ nm is lattice constant of monolayer graphene, respectively. The second line in Eq. (1) describes the moiré tunneling between the two graphene layers. Such tunneling is encoded in the matrix

$$T_j^{\xi} = w_0 - w_1 e^{i\xi(2\pi/3)j\sigma_z} \sigma_x e^{-i\xi(2\pi/3)j\sigma_z}, \tag{2}$$

with the momenta $\mathbf{q}_0 = R_{-\theta/2}\mathbf{K}_+ - R_{\theta/2}\mathbf{K}_+$, $\mathbf{q}_1 = R_{2\pi/3}\mathbf{q}_0$ and $\mathbf{q}_2 = R_{-2\pi/3}\mathbf{q}_0$ (Fig. 1). $w_0$ and $w_1$ in $T_j^{\xi}$ are the tunneling strengths between $AA$ and $AB$ sites, respectively. *Ab initio* numerics gives $w_1 \approx 110$ meV [52], however, some works suggested smaller values [18, 21, 53, 54]. In this paper, we consider two situations with $w_1 = 90$ meV and $w_1 = 110$ meV to account for variations of different theoretical models and realistic samples. Furthermore, we fix $w_0 = 0.7w_1$ to include the effects of lattice relaxation [55, 56] and corrugation [54, 57, 58]. The two valleys are decoupled in $H_{\text{kin}}$ and can be related to each other by time-reversal conjugate. To calculate the band structure at momentum $\mathbf{k}_0$ in the MBZ of a specific valley, we express $\mathbf{k}$ in $H_{\text{kin}}$ as $\mathbf{k}_0 + m\mathbf{G}_1 + n\mathbf{G}_2$ with integers $m, n = -d, ..., d$, where $d$ is a suitably chosen cutoff, then diagonalize $H_{\text{kin}}$ with the fixed valley index. As there is no alignment with the hexagonal boron nitride (hBN) substrate, the Dirac band touching exists at the corners of the MBZ.

Note that $H_{\text{kin}}$ is independent of electron's spin, leading to a two-fold spin degeneracy for each moiré band. For static TBG, interactions are able to lift this spin degeneracy under suitable circumstances [10, 59–62]. We hence drop this spin degeneracy by assuming spin polarization throughout this work, which can significantly enhance numerical efficiency of the many-body simulation in Sec. 4. In fact, interactions can also induce valley polarization in static TBG. However, as shown in Sec. 2.2, the time-reversal symmetry between two valleys is broken when the system is driven by light. Therefore, light driving could lead to different physics in the two valleys from the static TBG. We will keep the valley degree of freedom to take this effect of light into account.

We simulate the interaction between electrons via the screened Coulomb potential

$$H_{\text{int}} = \frac{1}{2} \sum_{\mathbf{q}} V(\mathbf{q}) : \rho(\mathbf{q})\rho(-\mathbf{q}) :, \tag{3}$$

where $\rho(\mathbf{q})$ is the electron's density operator and $::$ means the normal order. As the two valleys are decoupled in the single-electron level, we have $\rho(\mathbf{q}) = \sum_{\xi} \rho_{\xi}(\mathbf{q})$, where $\rho_{\xi}(\mathbf{q})$ is the density operator of electrons in valley $\xi$. We choose the Yukawa potential $V(\mathbf{q}) = \frac{e^2}{4\pi\epsilon_r\epsilon_0 S} \frac{2\pi}{\sqrt{|\mathbf{q}|^2 + \kappa^2}}$

to describe the screening, where $e$ is the electron charge, $\epsilon_r$ is the relative dielectric constant of the material, $\epsilon_0$ is the dielectric constant of vacuum, $S$ is the area of the moiré superlattice, and $\kappa$ measures the screening strength. Throughout this work, we fix $\epsilon_r = 4$ [63, 64] and $\kappa = 1/a_M$, with $a_M = a/(2\sin(\theta/2))$ the lattice constant of TBG.

## 2.2 Floquet system

Now we consider the coupling of TBG with light in the scenario of periodic driving. We suppose that the system is driven by a circularly polarized light which shines vertically and uniformly across the surface of TBG. The light field is represented by an electric field rotating in-plane as $\mathbf{E} = \mathcal{E}_0 (\sin\Omega t, \cos\Omega t)$, where $\Omega$ is the driving frequency. The corresponding vector potential is $\mathbf{A} = A_0 (\cos\Omega t, -\sin\Omega t)$ with $A_0 = \mathcal{E}_0/\Omega$, satisfying $\mathbf{E} = -\frac{\partial \mathbf{A}}{\partial t}$. For the single-electron Hamiltonian, the light field only affects the intralayer hopping. This is because the interlayer tunneling is dominated by hopping between atoms that are exactly on top of each other, thus mostly contributed by $z$-component of the vector potential which is absent in our setup [36, 37, 39]. We include the effect of light using a Peierls substitution $\mathbf{k} \rightarrow \mathbf{k} + e\mathbf{A}(t)/\hbar$ in the intralayer terms of Eq. (1), resulting in a time-dependent single-particle Hamiltonian $H_{\text{kin}}(t)$. The interaction Hamiltonian Eq. (3) remains as in the static case since it has the density-density form [48, 65]. Combining both terms, we get a new time-periodic Hamiltonian

$$H(t) = H_{\text{kin}}(t) + H_{\text{int}}, \tag{4}$$

to describe our system irradiated by the circularly polarized light, where $H(t) = H(t + 2\pi/\Omega)$.

According to the Floquet theory, the stroboscopic evolution of the system, upon a unitary transformation, can be captured by an effective static Hamiltonian $H_{\text{eff}}$ that does not depend on initial conditions [31–33]. While in general it is complicated to evaluate $H_{\text{eff}}$, we consider the limit where the driving frequency $\Omega$ is large compared to other characteristic energy scales in the system. In this case, $H_{\text{eff}}$ can be represented by a series expansion of $1/\Omega$ [31–33,49,65]:

$$H_{\text{eff}} = H_{\text{eff}}^{(0)} + H_{\text{eff}}^{(1)} + H_{\text{eff}}^{(2)} + \dots, \tag{5}$$

with

$$H_{\text{eff}}^{(0)} = H_0, \tag{6a}$$

$$H_{\text{eff}}^{(1)} = \frac{1}{\hbar\Omega} \sum_{m=1}^{\infty} \frac{[H_m, H_{-m}]}{m}, \tag{6b}$$

$$H_{\text{eff}}^{(2)} = \frac{1}{(\hbar\Omega)^2} \left\{ \sum_{m=1}^{\infty} \frac{[H_m, [H_0, H_{-m}]]}{2m^2} + \sum_{\substack{m,m'=1 \\ m \neq m'}}^{\infty} \frac{[H_{-m'}, [H_{m'-m}, H_m]] - [H_{m'}, [H_{-m'-m}, H_m]]}{3mm'} \right\}$$

$$+ \text{H.c.} \tag{6c}$$

Here $H_m$ is the Fourier transform of $H(t)$, i.e., $H(t) = \sum_m H_m e^{im\Omega t}$. For our model, $H_m$ is nonzero only when $m = 0, \pm 1$.

The zeroth-order term $H_{\text{eff}}^{(0)} = H_0$ is just the static Hamiltonian $H_{\text{kin}} + H_{\text{int}}$. The first-order term is

$$H_{\text{eff}}^{(1)} = \frac{(eA_0 v_F)^2}{\hbar\Omega} \sum_{\xi=\pm} \sum_{\mathbf{k}} \xi \left( \psi_{t,\xi}^{\dagger}(\mathbf{k}) \sigma_z \psi_{t,\xi}(\mathbf{k}) + \psi_{b,\xi}^{\dagger}(\mathbf{k}) \sigma_z \psi_{b,\xi}(\mathbf{k}) \right), \tag{7}$$

which is the same as that derived for the non-interacting TBG [36–38, 44] because the interaction is time-independent in our model. $H_{\text{eff}}^{(1)}$ is a single-particle term which introduces a

staggered potential of strength $P = (eA_0 v_F)^2/(\hbar\Omega) = (3a^2 t_0^2 e^2 \mathcal{E}_0^2)/(4\hbar^3\Omega^3)$ in both graphene layers. Note that this potential is opposite in the two valleys.

Previous works studying the high-frequency driving in non-interacting lattice models often neglect $H_{\text{eff}}^{(2)}$ and other higher-order terms in $H_{\text{eff}}$, because their corrections to the single-particle Hamiltonian is quite small. However, once interactions are considered, one should be very careful when dealing with these high-order terms, because they can include effective interactions even though the original interaction $H_{\text{int}}$ is time-independent. To the leading order, these effective interactions are present in $H_{\text{eff}}^{(2)}$ if $[H_m, [H_{\text{int}}, H_{-m}]] \neq 0$. While being much weaker than $H_{\text{eff}}^{(0)}$ and $H_{\text{eff}}^{(1)}$, these effective interactions may still be comparable to the many-body gap protecting the ground state of $H_{\text{eff}}$, thus having essential influences on the low-energy stroboscopic physics. Indeed, it was found in some Floquet topological lattice models that the effective interactions in $H_{\text{eff}}^{(2)}$ led by original density-density repulsions destabilize topologically ordered FCIs [49]. In our model, we carefully evaluate $[H_1, [H_{\text{int}}, H_{-1}]]$. Remarkably, we find it is zero due to the special forms of $H_{\pm 1}$ and $H_{\text{int}}$ in our model (see Appendix A). Therefore, by contrast to Ref. [49], $H_{\text{eff}}^{(2)}$ in our model is still a single-particle correction without effective interactions:

$$H_{\text{eff}}^{(2)} = \frac{(eA_0 v_F)^2}{(\hbar\Omega)^2}\left[ -H_{\text{kin}} + \sum_{\xi=\pm}\sum_{\mathbf{k}}\sum_{j=0}^{2}\left(\psi_{t,\xi}^\dagger\left(\mathbf{k} - \xi\mathbf{q}_0 + \xi\mathbf{q}_j\right)\mathbb{W}_\theta^\xi \psi_{b,\xi}(\mathbf{k}) + \text{H.c.}\right)\right], \quad (8)$$

where

$$\mathbb{W}_\theta^\xi = w_0 \begin{pmatrix} e^{i\xi\theta} & 0 \\ 0 & e^{-i\xi\theta} \end{pmatrix}. \quad (9)$$

The periodic driving of light changes the symmetry of the single-particle TBG Hamiltonian. In the static case, $H_{\text{kin}}$ has the 180-degree in-plane rotation symmetry $\mathcal{C}_2$ and the time-reversal symmetry $\mathcal{T}$ [66], whose actions on the spinors of electron operators are

$$\mathcal{C}_2 \psi_{t/b,\xi}(\mathbf{k})\mathcal{C}_2^{-1} = \tau_x \sigma_x \psi_{t/b,\xi}(-\mathbf{k}), \ \mathcal{T}\psi_{t/b,\xi}(\mathbf{k})\mathcal{T}^{-1} = \tau_x \psi_{t/b,\xi}(-\mathbf{k}). \quad (10)$$

Here $\tau_{x,y,z}$ are the Pauli matrices acting on the valley degree of freedom. In the presence of light driving, both $\mathcal{C}_2$ and $\mathcal{T}$ symmetries are preserved by $H_{\text{eff}}^{(0)}$ and $H_{\text{eff}}^{(2)}$. However, $H_{\text{eff}}^{(1)}$ contains $\tau_z \sigma_z$ and behaves like a Haldane mass term for Dirac fermions, so it breaks the $\mathcal{T}$ symmetry. As a result, the band gap at the Dirac points can be opened, leading to isolated valence and conduction bands near the CNP in each valley. These valence and conduction bands can carry non-zero Chern numbers in specific range of $P$ and twist angle $\theta$ [36,38]. Moreover, the Chern numbers of the valence (or conduction) bands in opposite valleys are expected to be the same because the Dirac fermions gain opposite masses. While the $\mathcal{T}$ symmetry is broken by light driving, the $\mathcal{C}_2$ symmetry survives, which makes the band structure of the driven TBG invariant under $\xi \to -\xi$ and $\mathbf{k} \to -\mathbf{k}$.

We would like to compare light driving with the alignment of static TBG with an hBN substrate, which can also lift the band touching at the Dirac points [14]. Unlike the light driving, the hBN alignment breaks $\mathcal{C}_2$ but not $\mathcal{T}$. In this case, while the band energies are also invariant under $\xi \to -\xi$ and $\mathbf{k} \to -\mathbf{k}$ due to the remaining $\mathcal{T}$ symmetry, the Chern numbers of the valence (or conduction) bands in the two valleys must be opposite. We will discuss this symmetry difference between light driving and hBN alignment more and explore its effect on the many-body physics in Sec. 4.2 and Appendix C.

In the following, we choose a high frequency $\hbar\Omega = 1.5$ eV, which significantly exceeds the energy scale of low-energy bands in static TBG. When $\Omega$ is fixed, $P$ quantifies the driving strength. We consider $P$ up to 60 meV, corresponding to a strong electric field $\mathcal{E}_0 \approx 8$ MV/cm. Then the prefactor $\frac{(eA_0 v_F)^2}{(\hbar\Omega)^2}$ of $H_{\text{eff}}^{(2)}$ is only $\sim 4\%$ at most.

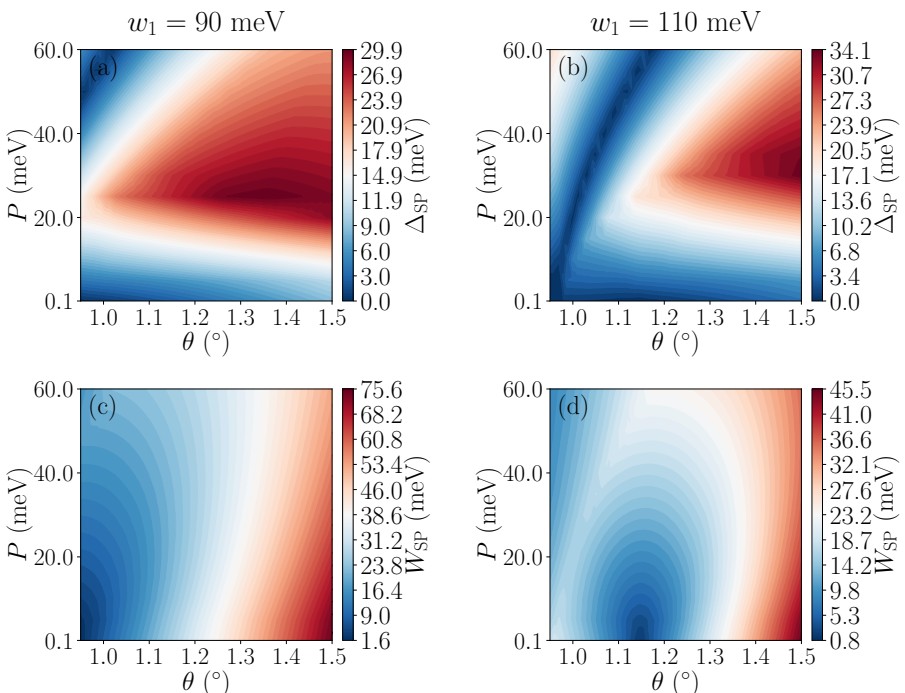

Figure 2: The indirect band gap $\Delta_{\text{SP}}$ and the bandwidth $W_{\text{SP}}$ of the Floquet valence band in a single valley for $w_1 = 90$ meV [(a),(c)] and $w_1 = 110$ meV [(b),(d)].

## 3  Floquet band structure

Now we have obtained the effective static Hamiltonian

$$H_{\text{eff}} = H_{\text{kin}} + H_{\text{int}} + H_{\text{eff}}^{(1)} + H_{\text{eff}}^{(2)}, \tag{11}$$

describing the stroboscopic nature of our Floquet system, which includes the original interaction and a single-particle part corrected by driving. Before we dive into the interaction induced many-body physics, let us first analyze the properties of the Floquet bands. To be concrete, we focus on the Floquet valence bands below the CNP. Because the bands in opposite valleys carry the same Chern number and their dispersion can be related by the $\mathbf{k} \to -\mathbf{k}$ transformation, we only need to consider a single valley.

In Fig. 2, we present the indirect band gap $\Delta_{\text{SP}}$ and bandwidth $W_{\text{SP}}$ of the valence band in a single valley as functions of $\theta$ and $P$. For most parameters that we consider, we find positive $\Delta_{\text{SP}}$, meaning that the Floquet valence band is isolated from other bands at these parameters. However, there are some lines in the $(\theta, P)$ space along which the band gap vanishes. After calculating the Chern number $C$ of the Floquet valance band, we find these gap-vanishing lines correspond to the transition between $C = -1$ and $C = 0$ (Fig. 3). In the parameter range that we consider, the largest band gaps appear in the topological $C = -1$ case.

## 4  Many-body physics

Because the Floquet valence bands in both valleys are isolated and carry Chern number $C = -1$ in a wide range of parameters $\theta$ and $P$ (Figs. 2 and 3), they are promising to host Floquet FCIs. Now we consider the situation in which the two Floquet valence bands, one in each valley, are partially filled by $N$ interacting electrons to examine this possibility. We choose a finite periodic

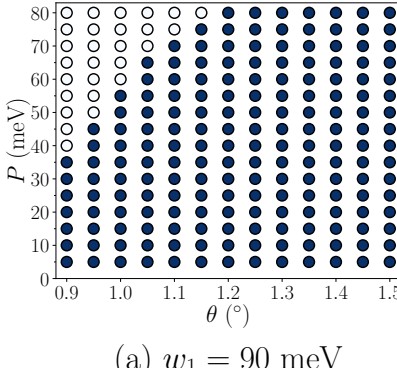
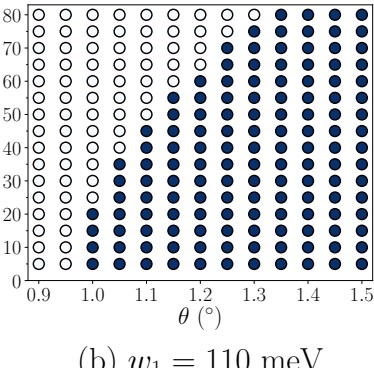

$$\text{(a) } w_1 = 90 \text{ meV} \qquad\qquad \text{(b) } w_1 = 110 \text{ meV}$$

Figure 3: Chern number of the Floquet valence band in a single valley for (a) $w_1 = 90$ meV and (b) $w_1 = 110$ meV. Solid dots represent $C = -1$, and circles represent $C = 0$.

system with $N_1$ and $N_2$ moiré unit cells in the directions of the two primitive moiré lattice vectors. The total filling $\nu$ in the two Floquet valence bands is defined as $N/(N_1 N_2)$. Due to the periodic boundary condition, each energy level of this finite system can be labeled by the total two-dimensional (2D) momentum $\mathbf{K} = (K_1, K_2)$, with integers $K_1 = 0, 1, \cdots, N_1 - 1$ and $K_2 = 0, 1, \cdots, N_2 - 1$. Motivated by the observations of robust FCIs at $\nu = 1/3$ (lattice analogs of the celebrated Laughlin state [67]) in various static $|C| = 1$ topological flat bands [68, 69], we fix $\nu = 1/3$ in our Floquet system.

We focus on the $C = -1$ region in the $(\theta, P)$ space to numerically study the effective static Hamiltonian Eq. (11) to search for the evidence of Floquet FCIs. Since the Floquet valence bands are well isolated in the $C = -1$ region, it is fair to project Eq. (11) to these active bands, leading to

$$H_{\text{eff}}^{\text{proj}} = \sum_{\mathbf{k} \in \text{MBZ}} \sum_{\xi = \pm} E_\xi(\mathbf{k}) c_{\mathbf{k},\xi}^\dagger c_{\mathbf{k},\xi} + \sum_{\{\mathbf{k}_i\} \in \text{MBZ}} \sum_{\xi,\xi' = \pm} V_{\mathbf{k}_1 \mathbf{k}_2 \mathbf{k}_3 \mathbf{k}_4}^{\xi,\xi'} c_{\mathbf{k}_1,\xi}^\dagger c_{\mathbf{k}_2,\xi'}^\dagger c_{\mathbf{k}_3,\xi'} c_{\mathbf{k}_4,\xi}, \qquad (12)$$

where $c_{\mathbf{k},\xi}^\dagger$ ($c_{\mathbf{k},\xi}$) is the operator creating (annihilating) an electron with momentum $\mathbf{k} \in \text{MBZ}$ in the Floquet valence band of valley $\xi$, $E_\xi(\mathbf{k})$ is the corresponding band dispersion, and the matrix element $V_{\mathbf{k}_1 \mathbf{k}_2 \mathbf{k}_3 \mathbf{k}_4}^{\xi,\xi'}$ is given by [18]

$$V_{\mathbf{k}_1 \mathbf{k}_2 \mathbf{k}_3 \mathbf{k}_4}^{\xi,\xi'} = \frac{1}{2} \delta'_{\mathbf{k}_1 + \mathbf{k}_2, \mathbf{k}_3 + \mathbf{k}_4} \sum_{\mathbf{G}} V(\mathbf{k}_1 - \mathbf{k}_4 + \mathbf{G}) \left( \langle u_\xi(\mathbf{k}_1) | u_\xi(\mathbf{k}_4 - \mathbf{G}) \rangle \langle u_{\xi'}(\mathbf{k}_2) | u_{\xi'}(\mathbf{k}_3 + \mathbf{G} + \delta\mathbf{G}) \rangle \right). \qquad (13)$$

Here $\delta'_{\mathbf{k},\mathbf{k}'}$ is the 2D periodic Kronecker delta function with the period of MBZ reciprocal lattice vectors, $|u_\xi(\mathbf{k})\rangle$ is the Floquet valence band eigenvector in valley $\xi$, $\delta\mathbf{G} = \mathbf{k}_1 + \mathbf{k}_2 - \mathbf{k}_3 - \mathbf{k}_4$, and the sum of $\mathbf{G}$ is over the entire reciprocal space rather than only in a single MBZ. $E_\xi(\mathbf{k})$ and $|u_\xi(\mathbf{k})\rangle$ can be obtained by diagonalizing the single-particle part of $H_{\text{eff}}$ with a fixed valley index $\xi$.

The number of electrons $N_\xi$ in each valley $\xi$ is preserved in the projected effective Hamiltonian Eq. (12), which allows us to define a $z$-direction pseudospin $S_z = (N_+ - N_-)/2$. For a given number of electrons $N$, $S_z$ varies from $-N/2$ to $N/2$. We then use exact diagonalization to extract the low-energy physics of Eq. (12) in each $S_z$ sector. Due to the $\mathcal{C}_2$ symmetry which is not broken by the light driving, the energy spectrum of Eq. (12) should be invariant under $S_z \to -S_z$ and $\mathbf{K} \to -\mathbf{K}$. We have examined that the ground-state energy gap obtained from diagonalizing Eq. (12) is always smaller than the single-electron band gaps of the Floquet valence bands, thus justifying the validity of band projection.

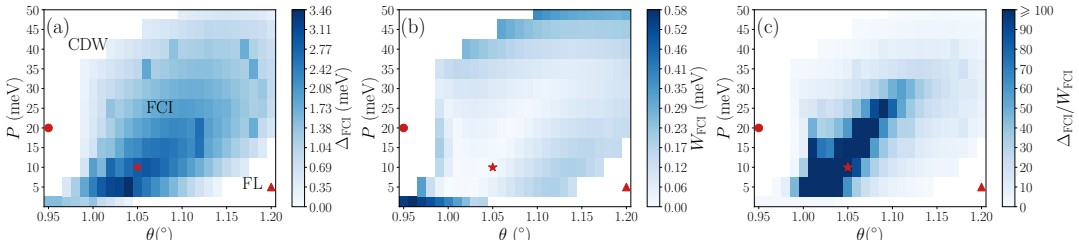

Figure 4: (a) The FCI gap $\Delta_{\mathrm{FCI}}$, (b) FCI splitting $W_{\mathrm{FCI}}$, and (c) their ratio $\Delta_{\mathrm{FCI}}/W_{\mathrm{FCI}}$ for $N = 10$ electrons in the valley $\xi = +$. The number of unit cells is $N_1 \times N_2 = 5 \times 6$. We choose $w_1 = 90$ meV. The three-fold FCI degeneracy is absent in white regions. In (a), we give the tentative phase diagram in a single valley. As shown later, the CDW phase is identified by the structure factor, and the FL phase is characterized by the step structure in the $n(\mathbf{k}) - E_h(\mathbf{k})$ curve. The markers indicate the representative parameter points that we choose in Figs. 5 and 6.

## 4.1 Many-body physics in a single valley

Let us first assume valley polarization and study the many-body physics in a single valley. Because the two valleys are related by the $\mathcal{C}_2$ symmetry, we can just use $\xi = +$ as a representative valley, namely, we consider the $S_z = N/2$ sector in Eq. (12). Such valley polarization reduces the dimension of Hilbert space, thus significantly increasing numerical efficiency. In what follows, we choose $w_1 = 90$ meV. As shown in Appendix B, the results of $w_1 = 110$ meV are similar.

On the torus geometry, an essential feature of the $\nu = 1/3$ Laughlin FCI is the robust three-fold ground-state degeneracy in momentum sectors determined by the Haldane statistics of the $\nu = 1/3$ Laughlin state [68,70,71]. Therefore, we compute the FCI gap $\Delta_{\mathrm{FCI}}$ in the $C = -1$ region as the energy difference between the fourth and the first eigenvalues of the projected effective Hamiltonian Eq. (12) in valley $\xi = +$, where all eigenvalues are sorted in ascending order. If the lowest three eigenstates are not in momentum sectors predicted by the Haldane statistics, we simply set the FCI gap to be zero. Meanwhile, we also measure the FCI splitting $W_{\mathrm{FCI}}$, quantified by the energy difference between the third and the first eigenvalues when they are in the FCI momentum sectors. The result for $N = 10$ electrons is demonstrated in Figs. 4(a) and 4(b). Strikingly, we can identify a wide range of parameters for which the lowest three eigenstates are located in FCI momentum sectors, protected by a significant gap, and approximately degenerate [i.e., very large $\Delta_{\mathrm{FCI}}/W_{\mathrm{FCI}}$, as shown in Fig. 4(c)].

To examine the robustness of such ground-state degeneracies, we examine their dependence on the system size and the boundary condition. In Fig. 5(a), we demonstrate the low-energy spectra of Eq. (12) in the valley $\xi = +$ for $N = 8$, 10, and 12 electrons at a representative parameter point $(\theta, P) = (1.05°, 10 \text{ meV})$ [labelled by the stars in Figs. 4(a)-4(c)]. For each system size, we observe excellent three-fold ground-state degeneracy, i.e., the splitting of the three ground states is much smaller than their separation from higher-energy levels. The finite-size scaling of the ground-state splitting and the energy gap suggests that both the three-fold degeneracy and the ground-state gap are very likely to survive in the thermodynamic limit [Fig. 5(b)]. By inserting magnetic flux through the handles of the toroidal system, we find that the three-fold ground-state degeneracy persists [Fig. 5(c)]. All these data further confirm the robustness of the three-fold topological degeneracy. Remarkably, the observed gap corresponds to a temperature of about 20 Kelvin, which is an order of magnitude higher than required by the conventional fractional quantum Hall states in 2D electron gases. It is often useful to express this energy gap in terms of the Coulomb interaction strength in the system, which we estimate as $U = e^2/(4\pi\epsilon_0\epsilon_r a_M)$. We find $U$ is about 26.8 meV at the parameter


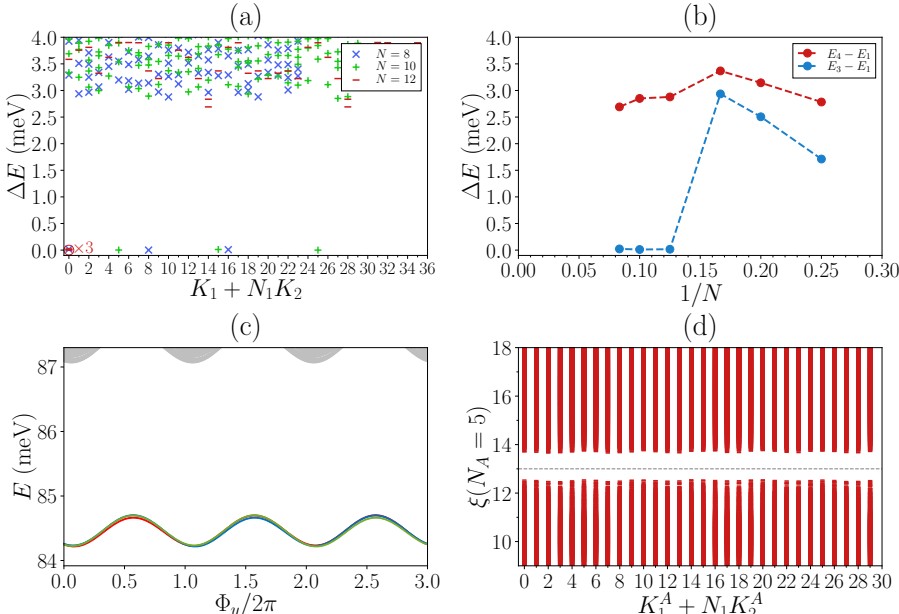

Figure 5: The $\nu = 1/3$ Laughlin FCI in the Floquet valence band of valley $\xi = +$ at $\theta = 1.05°$, $P = 10$ meV when $w_1 = 90$ meV. (a) The low-lying energy spectra for $N = 8, 10, 12$ electrons on the $N/2 \times 6$ lattice. (b) The finite-size scaling of the energy gap and the ground-state splitting for $N = 4, 5, 6, 8, 10, 12$ electrons. (c) The spectral flow for $N = 10$, $N_1 \times N_2 = 5 \times 6$, where $\Phi_y$ is the magnetic flux insertion in the $\mathbf{a}_2$ direction. (d) The particle entanglement spectrum for $N = 10$, $N_1 \times N_2 = 5 \times 6$ and $N_A = 5$, with 23256 levels below the entanglement gap (the dashed line).

point in Fig. 5, so the energy gap is about $0.07U$. Because we include the band dispersion in the numerical simulation, the many-body gap is not simply proportional to $U$. Moreover, this energy gap in general changes with the filling factor and the choice of active bands (valence versus conduction bands). It can be further reduced by the intervalley excitations (see Sec. 4.2), spinful excitations, and disorder.

To further corroborate the nontrivial topological properties of the ground state, we also investigate the particle entanglement spectrum (PES), which encodes the information of quasihole excitations of the system and distinguishes FCIs from other competing phases [68, 72]. In Fig. 5(d), we divide the whole system into $N_A$ and $N - N_A$ electrons and label each PES level by the total momentum $(K_1^A, K_2^A)$ of those $N_A$ electrons. A clear entanglement gap appears separating the low-lying PES levels from higher ones, and the number of levels below the gap exactly matches the pertinent counting of quasihole excitations in the $\nu = 1/3$ Laughlin state [68, 70, 71]. This entanglement spectroscopy, together with the low-energy spectrum, strongly suggests that in a single valley the most robust $\nu = 1/3$ Floquet Laughlin FCI exists in the region with $\theta \approx 1.0° - 1.1°$ and $P \approx 5$ meV $- 30$ meV when $w_1 = 90$ meV.

There are also regions in Fig. 4(a) in which the three-fold topological degeneracy of the ground states becomes poor and eventually collapses. On the left side of the FCI phase (with smaller $\theta$), we find a pronounced sensitivity of the energy spectrum to the lattice size. For $N = 6$, $N_1 \times N_2 = 3 \times 6$ and $N = 12$, $N_1 \times N_2 = 6 \times 6$, we observe a new kind of three-fold ground-state degeneracy in different momentum sectors from the $\nu = 1/3$ Laughlin FCI. These momentum sectors are separated exactly by the moiré Dirac point momenta $\mathbf{K}_+^b$ and $\mathbf{K}_+^t$. For example, while the three Laughlin FCI states of $N = 12$, $N_1 \times N_2 = 6 \times 6$ all carry $(K_1, K_2) = (0, 0)$ [Fig. 5(a)], the three ground states at the parameter point $(\theta, P) = (0.95°, 20$ meV) [labelled by the dot in Figs. 4(a)-4(c)] of the same system size are located in the $(K_1, K_2) = (0, 0), (2, 2)$

and (4, 4) sectors [Fig. 6(a)], which are separated by momentum $\Delta\mathbf{K} = (2,2) = (\mathbf{G}_1 + \mathbf{G}_2)/3 \sim \mathbf{K}_+^b$ and $\Delta\mathbf{K} = (4,4) = 2(\mathbf{G}_1 + \mathbf{G}_2)/3 \sim \mathbf{K}_+^t$ (Fig. 1). Here $\sim$ means "equal to" up to a MBZ reciprocal lattice vector. The distribution of degenerate ground states over equally spaced momenta is a signal of charge density waves. To further confirm this, we compute the structure factor $S(\mathbf{q})$ which can reveal the CDW order. We define $S(\mathbf{q})$ in a single valley as

$$S(\mathbf{q}) = \frac{1}{N_1 N_2} \left( \langle \bar{\rho}_\xi(\mathbf{q}) \bar{\rho}_\xi(-\mathbf{q}) \rangle - N^2 \delta_{\mathbf{q},0} \right), \tag{14}$$

where $\bar{\rho}_\xi(\mathbf{q}) = \sum_{\mathbf{k} \in \mathrm{MBZ}} \langle u_\xi(\mathbf{k}) | u_\xi(\mathbf{k} - \mathbf{q}) \rangle c_{\mathbf{k},\xi}^\dagger c_{\mathbf{k}-\mathbf{q},\xi}$ is the density operator projected to the Floquet valence band in valley $\xi$. Remarkably, we find pronounced peaks at the corners of the MBZ [Figs. 6(b) and 6(c)], revealing the underlying phase is the CDW with the order momentum $\mathbf{K}_+^{t,b}$. Various types of CDW phases have been identified in static TBG-hBN as competing phases against FCIs [21,73–75]. The $\mathbf{K}$-CDW states corresponds to a Wigner crystal, whose unit cell is tripled compared to the original moiré lattice [21]. On the other hand, we do not see the CDW degeneracy among the lowest states for $N = 8, N_1 \times N_2 = 4 \times 6$ and $N = 10, N_1 \times N_2 = 5 \times 6$. This is because the $\mathbf{K}_+^{t,b}$ points are absent in the MBZ of these finite lattices. Remember that the single-electron momentum $\mathbf{k}$ can only take $\mathbf{k} = \frac{m_1}{N_1}\mathbf{G}_1 + \frac{m_2}{N_2}\mathbf{G}_2$ for a finite periodic system. Therefore, both $N_1$ and $N_2$ must be divisible by three if $\mathbf{K}_+^{t,b}$ belong to this set of allowed $\mathbf{k}$. Otherwise, the $\mathbf{K}$-CDW cannot develop in the finite system.

Finally, we go to the region on the right side of the FCI phase (larger $\theta$), where we see neither the FCI topological degeneracy nor the CDW degeneracy in the low-energy spectra. As shown in Fig. 2(c), the bandwidth in this region becomes larger than that in the FCI region, so that the stronger band dispersion may dominate over the interaction, leading to a Fermi liquid state. We confirm this by studying the correlation between the electron's ground-state occupation $\langle n(\mathbf{k}) \rangle$ at momentum $\mathbf{k}$ and the energy $E(\mathbf{k})$ of the Floquet valence band in valley $\xi = +$. The result at a representative parameter point [labelled by the triangles in Figs. 4(a)-4(c)] is displayed in Fig. 6(d). We observe a striking Fermi surface-like structure in the $\langle n(\mathbf{k}) \rangle - E(\mathbf{k})$ data, that is, $\langle n(\mathbf{k}) \rangle \approx 1$ for small $E(\mathbf{k})$ and suddenly drops for larger $E(\mathbf{k})$. This feature strongly suggests that the region on the right side of the FCI phase is a Fermi liquid phase dominated by the band dispersion.

Based on numerical results above, we present a tentative phase diagram in Fig. 4(a) for the many-body physics in the $\nu = 1/3$ filled Floquet valence band of a single valley with $w_1 = 90$ meV. Because of the limited system sizes which we can reach by exact diagonalization, the phase boundaries are only roughly determined by the topological degeneracy of FCIs for 10 electrons, so they should not be thought as being precise.

It is well known that the quantum geometry of a flat band [76–78], such as the Berry curvature $\mathcal{B}(\mathbf{k})$ and the Fubini-Study (FS) metric $g(\mathbf{k})$, plays a crucial role in determining the many-body physics occurring in the band. Given the band eigenvector $|u(\mathbf{k})\rangle$, the FS metric and the Berry curvature are the real and imaginary part of the quantum geometric tensor $\mathcal{Q}(\mathbf{k})$, respectively:

$$\mathcal{Q}^{ab}(\mathbf{k}) = \langle \partial_{\mathbf{k}}^a u(\mathbf{k}) | \partial_{\mathbf{k}}^b u(\mathbf{k}) \rangle - \langle \partial_{\mathbf{k}}^a u(\mathbf{k}) | u(\mathbf{k}) \rangle \langle u(\mathbf{k}) | \partial_{\mathbf{k}}^b u(\mathbf{k}) \rangle \equiv g^{ab}(\mathbf{k}) - \frac{i}{2}\mathcal{B}^{ab}(\mathbf{k}), \tag{15}$$

where $a, b = x, y$ and $\mathcal{B}^{xy}(\mathbf{k}) = \mathcal{B}(\mathbf{k})$. It can be proved for a 2D band that $\mathrm{tr}g(\mathbf{k}) \geq |\mathcal{B}(\mathbf{k})|$ [16, 77, 79, 80]. FCI states are expected to be stable when the Berry curvature does not strongly fluctuate in the MBZ and the trace condition $\mathrm{tr}g(\mathbf{k}) = |\mathcal{B}(\mathbf{k})|$ is only weakly violated [19, 21, 22, 74, 75, 79–82]. To see this in our model, we compute the Berry curvature fluctuation and the violation of the trace condition, as quantified by

$$\sigma(\mathcal{B}) = \sqrt{\langle \mathcal{B}^2(\mathbf{k}) \rangle - \langle \mathcal{B}(\mathbf{k}) \rangle^2},$$
$$\delta_{\mathrm{tr}} = \langle \mathrm{tr}g^{ab}(\mathbf{k}) - |\mathcal{B}(\mathbf{k})| \rangle, \tag{16}$$

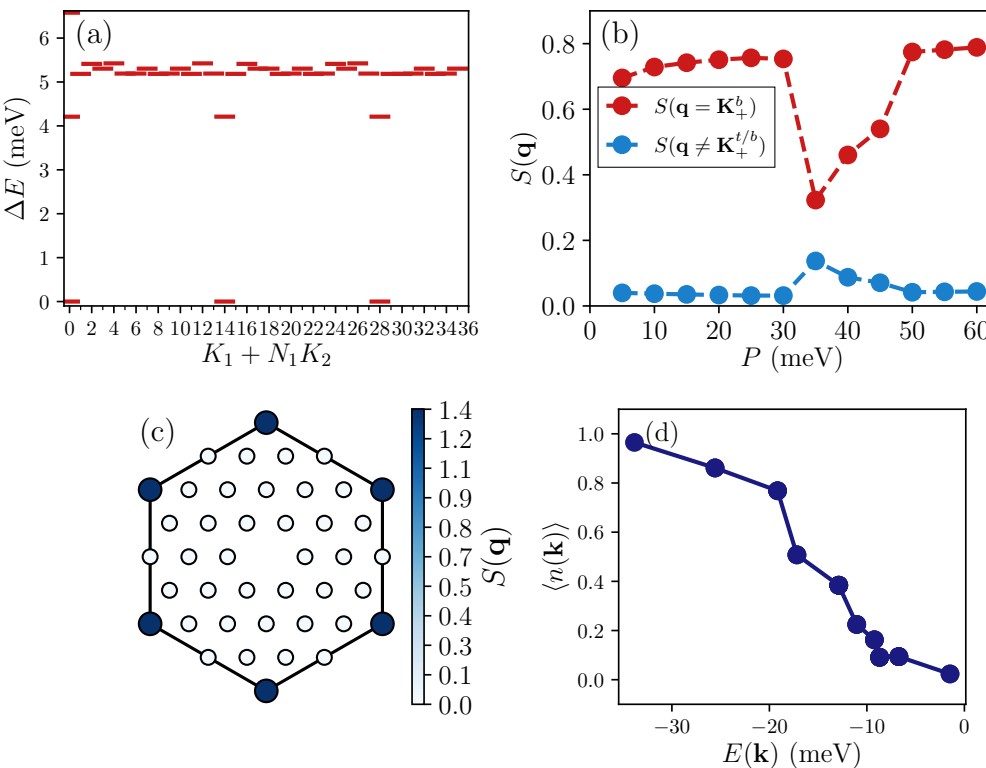

Figure 6: The competing phases of FCIs in the Floquet valence band of valley $\xi = +$ when $w_1 = 90$ meV. (a) The many-body energy spectrum at $(\theta, P) = (0.95°, 20\text{ meV})$ for $N = 12$, $N_1 \times N_2 = 6 \times 6$. The lowest three states are located at $(K_1, K_2) = (0,0), (2,2), (4,4)$, separated by $\Delta\mathbf{K} = (2,2)$ and $\Delta\mathbf{K} = (4,4)$. (b) The structure factor at $\theta = 0.95°$ as a function of $P$ for $N = 6$, $N_1 \times N_2 = 3 \times 6$. The values of $S(\mathbf{q})$ at $\mathbf{q} = \mathbf{K}_+^{t,b}$ are significantly larger than those at $\mathbf{q} \neq \mathbf{K}_+^{t,b}$. (c) Distribution of $S(\mathbf{q})$ in the MBZ for $N = 12$, $N_1 \times N_2 = 6 \times 6$ at the same parameter point with (a). (d) The ground-state occupation $\langle n(\mathbf{k}) \rangle$ at momentum $\mathbf{k}$ as a function of band energy $E(\mathbf{k})$ at $(\theta, P) = (1.2°, 5\text{ meV})$ for $N = 12$, $N_1 \times N_2 = 6 \times 6$.

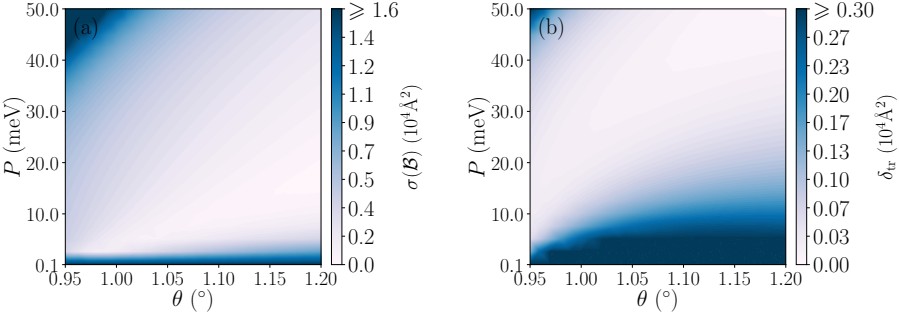

Figure 7: Quantum geometry of the Floquet valence band in a single valley for $w_1 = 90$ meV. (a) The fluctuation of the Berry curvature in the $(\theta, P)$ parameter space. (b) The violation of the trace condition in the $(\theta, P)$ parameter space.

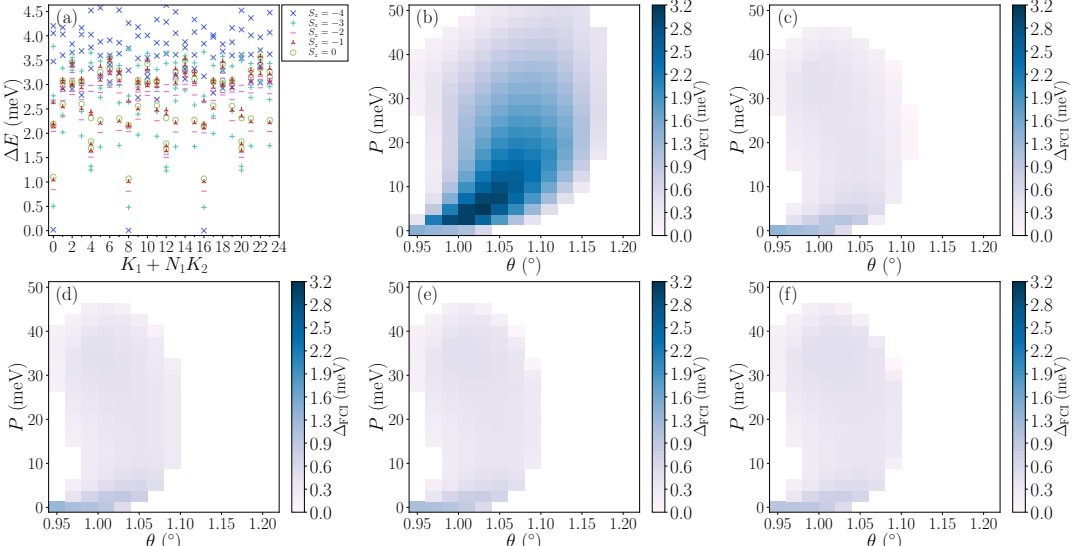

Figure 8: The many-body physics in Floquet valence bands when both valleys are considered. We choose $w_1 = 90$ meV and focus on the system size $N = 8, N_1 \times N_2 = 4 \times 6$. (a) The low-energy spectra in the $S_z = 0, -1, -2, -3, -4$ sectors at $(\theta, P) = (1.05°, 5\text{ meV})$. The spectrum in the $-S_z$ sector can be obtained by changing $\mathbf{K}$ to $-\mathbf{K}$. The Laughlin gaps in the $(\theta, P)$ parameter space are shown for (b) $S_z = \pm 4$, (c) $S_z = \pm 3$, (d) $S_z = \pm 2$, (e) $S_z = \pm 1$ and (f) $S_z = 0$.

respectively, for the Floquet valence band in a single valley, where $\langle O(\mathbf{k}) \rangle = \frac{1}{A_{\text{MBZ}}} \int_{\text{MBZ}} O(\mathbf{k}) d\mathbf{k}$ is the average of quantity $O$ in the MBZ, with $A_{\text{MBZ}}$ the area of MBZ. We plot $\sigma(\mathcal{B})$ and $\delta_{\text{tr}}$ for various $(\theta, P)$ parameters in Fig. 7. After comparing with Fig. 4 and Fig. 2(c), we find the most robust FCI states indeed appear when both $\sigma(\mathcal{B})$ and $\delta_{\text{tr}}$, as well as the bandwidth, are sufficiently small. Nevertheless, the minima of $\sigma(\mathcal{B})$, $\delta_{\text{tr}}$, and the bandwidth are not located at the same parameter point, so over-minimizing an individual quantity does not necessarily improve the stability of the FCI phase.

## 4.2 Many-body physics in two valleys

Now we relax the assumption of valley polarization to explore the effect of valley degree of freedom on the many-body phase diagram. We still choose $w_1 = 90$ meV in this section. Because the Hilbert space becomes much larger in the absence of valley polarization, in the following we focus on $N = 8$ electrons for which numerical simulations can be done efficiently.

First, we examine the low-energy spectrum of Eq. (12) in each $S_z$ sector. Remarkably, for each $S_z$ sector we can find a region in the $(\theta, P)$ parameter space where nice three-fold ground-state degeneracies with Laughlin momenta exists, as shown in Fig. 8. This feature originates from the $w_0 = 0$ limit, in which our driven system has the SU(2) symmetry among the two $C = -1$ Floquet valence bands in the two valleys. This SU(2) symmetry was at first noticed in the static TBG system [66]. We have numerically examined that each many-body energy level in the $w_0 = 0$ limit is exactly $(2S + 1)$-fold degenerate, where $S = 0, 1, \cdots, N/2$ is the total pseudospin quantum number (compared to $S_z$ which is the $z$-component pseudospin quantum number). In that case, the ground states in the FCI phase are FCI ferromagnetism with the largest $S = N/2$, and the Laughlin FCI states in different $S_z$ sectors should be related by pseudospin ladder operators. When $w_0$ is turned on, the $(2S + 1)$-fold degeneracy is lifted, so that the Laughlin FCIs in different $S_z$ sectors have different energies [Fig. 8(a)]. However, the Laughlin phase is preserved in each $S_z$ sector in a finite range of parameters. In Figs. 8(b)-

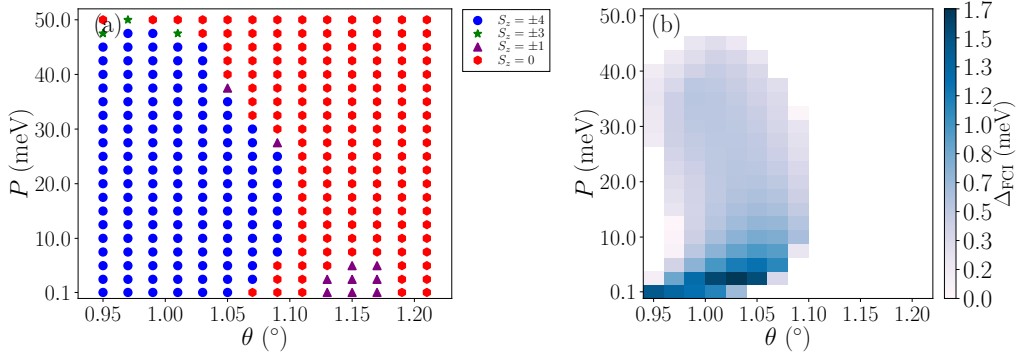

Figure 9: (a) The $S_z$ of the global ground state for $N = 8, N_1 \times N_2 = 4 \times 6$, $w_1 = 90$ meV. (b) The generalized FCI gap of the same system size.

8(f), we display the Laughlin gaps in all $S_z$ sectors. It can be seen that the Laughlin phase appears in similar regions of the parameter space. The valley-polarization sector ($S_z = \pm 4$) has the largest gap.

As the ground energies in different $S_z$ sectors can now be different, we plot the $S_z$ quantum number of the global ground state in Fig. 9(a). We find a large region in which the global ground state carries $S_z = \pm 4$, i.e., the ground state is valley polarized. After comparing with Fig. 4, we notice that this valley-polarized region almost covers the CDW phase and the strongest part of the FCI phase, so our identification of the phase diagram with the valley-polarization assumption holds in this region. Because the FCI states in this region can now exist in multiple $S_z$ sectors [Fig. 8(a)], we generalize our definition of the Laughlin gap to measure the energy difference between the ground state, if it is a Laughlin state in some $S_z$ sector, and the lowest excited state that is not a Laughlin state in some $S_z$ sector. This generalized FCI gap is shown in Fig. 9(b). Compared with Fig. 8(b), the gap is almost reduced by a factor of 2 due to intervalley excitations.

In Fig. 9(a), there is also a large $S_z = 0$ region on the right side of the valley-polarized region. Again, due to the larger bandwidth in this region [Fig. 2(c)], we expect that the band dispersion dominates over the interaction. To confirm this, we choose several parameter points in this region to compute the electron's ground-state occupation $\langle n(\mathbf{k}) \rangle$ at momentum $\mathbf{k}$. As shown in Fig. 10, we observe Fermi surface structures when plotting $\langle n(\mathbf{k}) \rangle$ as a function of the band energy. This means the ground state in the $S_z = 0$ region is a band-dispersion-induced Fermi liquid, in which electrons equally populate in the two valleys and occupy the momentum points from low band energies. Part of the FCI phase and the Fermi liquid phase in Fig. 4 is replaced by this $S_z = 0$ Fermi liquid phase once the valley-polarization assumption is relaxed.

## 5 Discussion

In this work, we investigate the interaction effect in twisted bilayer graphene irradiated with monochromatic circularly polarized light. We work in the regime of high driving frequency. When the Floquet valence bands obtained from the effective static Hamiltonian are partially occupied by electrons at total $\nu = 1/3$ filling, we find compelling numerical evidence that valley-polarized Floquet fractional Chern insulators exist for a wide range of twist angle and driving strength, as characterized by the robust ground-state topological degeneracy and the counting of quasihole excitations. By calculating the structure factor and the electron occupation numbers, the intriguing interplay of Floquet FCIs with charge density waves and Fermi liquid states are also identified in a single phase diagram. Our results demonstrate the rich many-body physics in Floquet TBG.

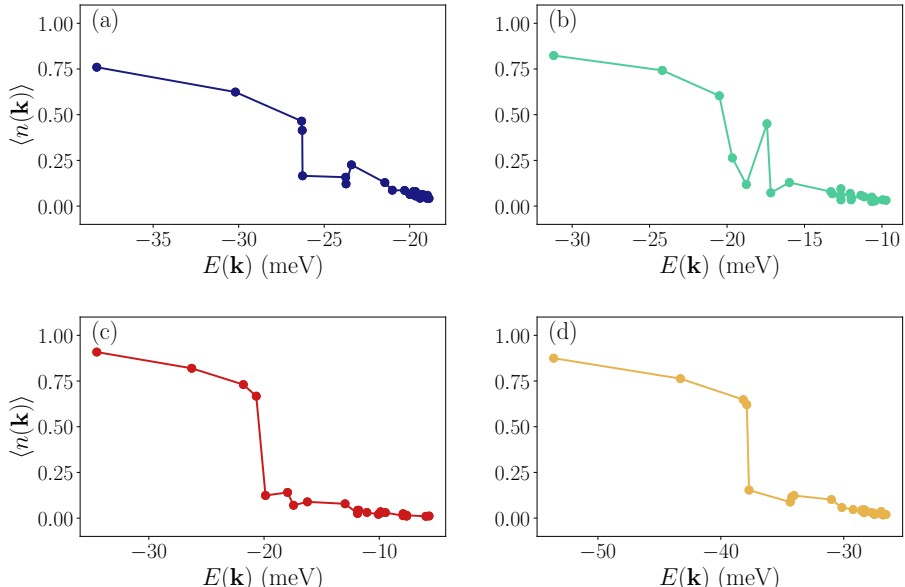

Figure 10: The ground-state occupation $\langle n(\mathbf{k}) \rangle$ at momentum $\mathbf{k}$ as a function of band energy $E(\mathbf{k})$ for $N = 8, N_1 \times N_2 = 4 \times 6$, $w_1 = 90$ meV in the $S_z = 0$ region. We choose four representative parameter points: (a) $(\theta, P) = (1.1°, 40 \text{ meV})$, (b) $(\theta, P) = (1.15°, 20 \text{ meV})$, (c) $(\theta, P) = (1.2°, 10 \text{ meV})$ and (d) $(\theta, P) = (1.2°, 50 \text{ meV})$.

There are several interesting theoretical future directions following our present work. First, it would be interesting to take into account more model parameters, such as the ratio $w_0/w_1$ and the dielectric constant, to explore the many-body phase diagram in a larger parameter space. The interlayer tunneling may be tuned by applying a longitudinal light. Second, while we only consider $\nu = 1/3$ filling in this work, it remains unclear whether FCIs at other fillings, especially non-Abelian ones, can be stabilized in this Floquet system. Finally, inspired by the recent development in Floquet band structures of other moiré materials beyond TBG [40–43], it is natural to study the Floquet many-body physics thereof.

Considering that the Floquet Chern insulator has been realized in experiments of mono-layer graphene [47], it is possible to observe it also in TBG driven by the light field, which is the first step towards realizing the Floquet FCIs predicted in our work. In fact, the parameters we choose here should be within experimentally realizable parameters. For example, both the driving frequency $\hbar\Omega = 1.5$ eV and the driving strength $\mathcal{E}_0 = 8$ MV/cm (corresponding to $P \approx 60$ meV) are accessible by current laser technology. However, obvious challenges still exist. For instance, the Floquet heating out of the long-lived prethermal regime will result a featureless infinite-temperature system. Fortunately, the time scale of the prethermal regime grows exponentially with the increasing driving frequency [83, 84]. Moreover, one should keep the driving off-resonant, otherwise the direct absorption of photons by electrons makes the effective static Hamiltonian insufficient to capture the out-of-equilibrium properties (such as the transport) of the system [24]. Finally, while our analysis of the low-energy physics of the effective Floquet Hamiltonian indicates that the driven TBG system can host Floquet FCIs, a deliberate time-evolution scheme is still needed to reach the desired states.

# Acknowledgments

**Funding information**   This project is supported by the National Key Research and Development Program of China through Grant No. 2020YFA0309200 and the National Natural Science Foundation of China through Grant No. 11974014. We thank the Tianhe-II platform at the National Supercomputer Center in Guangzhou for their allocation of CPU time.

# A   Effective Floquet Hamiltonian

Here we give a detailed derivation of the static effective Hamiltonian $H_{\text{eff}}$. Note that we do this derivation before the band projection. As discussed in the main text, the vertical light irradiation modifies the static single-particle TBG Hamiltonian $H_{\text{kin}}$ [Eq. (1)] by the Peierls substitution $\mathbf{k} \to \mathbf{k} + e\mathbf{A}(t)/\hbar$ in the intralayer part. The interlayer part in $H_{\text{kin}}$ [Eq. (1)] and the interaction $H_{\text{int}}$ [Eq. (3)] are not changed by the light field.

In the presence of light driving, the static Dirac Hamiltonian $h_{\pm\theta/2}^{\xi}\left(\mathbf{k} - \mathbf{K}_{\xi}^{t,b}\right)$ of each graphene layer of TBG becomes

$$-\hbar v_F \begin{pmatrix} 0 & [\xi\Delta k_{\xi,x}^{t,b} - i\Delta k_{\xi,y}^{t,b} + \xi\frac{eA_0}{\hbar}e^{i\xi\Omega t}]e^{\mp i\xi\theta/2} \\ [\xi\Delta k_{\xi,x}^{t,b} + i\Delta k_{\xi,y}^{t,b} + \xi\frac{eA_0}{\hbar}e^{-i\xi\Omega t}]e^{\pm i\xi\theta/2} & 0 \end{pmatrix}, \tag{A.1}$$

where $\xi = \pm$ is the valley index and $\Delta\mathbf{k}_{\xi}^{t,b} = \mathbf{k} - \mathbf{K}_{\xi}^{t,b}$. So the only non-zero Fourier components $H_m = \frac{1}{T}\int_0^T H(t)e^{-im\Omega t}dt$ of the total time-dependent Hamiltonian $H(t) = H_{\text{kin}}(t) + H_{\text{int}}$ are

$$H_0 = H_{\text{kin}} + H_{\text{int}},$$
$$H_1 = (-eA_0 v_F)\sum_{\mathbf{k}}\left(e^{i\theta/2}\psi_{tA,+}^{\dagger}(\mathbf{k})\psi_{tB,+}(\mathbf{k}) + e^{-i\theta/2}\psi_{bA,+}^{\dagger}(\mathbf{k})\psi_{bB,+}(\mathbf{k})\right)$$
$$+ (eA_0 v_F)\sum_{\mathbf{k}}\left(e^{i\theta/2}\psi_{tB,-}^{\dagger}(\mathbf{k})\psi_{tA,-}(\mathbf{k}) + e^{-i\theta/2}\psi_{bB,-}^{\dagger}(\mathbf{k})\psi_{bA,-}(\mathbf{k})\right),$$
$$H_{-1} = H_1^{\dagger}. \tag{A.2}$$

Using the relation $\left[c_m^{\dagger}c_n, c_k^{\dagger}c_l\right] = \delta_{n,k}c_m^{\dagger}c_l - \delta_{m,l}c_k^{\dagger}c_n$, we can obtain the first-order term of $H_{\text{eff}}$ as in Eq. (7).

The second-order term of $H_{\text{eff}}$ is

$$H_{\text{eff}}^{(2)} = \frac{1}{2(\hbar\Omega)^2}\left[H_1, [H_0, H_{-1}]\right] + \text{H.c.}$$
$$= \frac{1}{2(\hbar\Omega)^2}\left([H_1, [H_{\text{kin}}, H_{-1}]] + [H_1, [H_{\text{int}}, H_{-1}]]\right) + \text{H.c.} \tag{A.3}$$

A straightforward calculation gives $\frac{1}{2(\hbar\Omega)^2}\left[H_1, [H_{\text{kin}}, H_{-1}]\right]$ as in Eq. (8). The calculation of $[H_1, [H_{\text{int}}, H_{-1}]]$ is more tedious. We can write the interaction Eq. (3) in the second-quantized form:

$$H_{\text{int}} = \frac{1}{2}\sum_{\{\mathbf{k}_i\}}\sum_{\mathbf{q}}\sum_{\alpha,\beta}\sum_{\xi,\xi'}\delta'_{\mathbf{k}_1-\mathbf{k}_4,\mathbf{q}}\delta'_{\mathbf{k}_1+\mathbf{k}_2,\mathbf{k}_3+\mathbf{k}_4}V(\mathbf{q})\psi_{\alpha,\xi}^{\dagger}(\mathbf{k}_1)\psi_{\beta,\xi'}^{\dagger}(\mathbf{k}_2)\psi_{\beta,\xi'}(\mathbf{k}_3)\psi_{\alpha,\xi}(\mathbf{k}_4), \tag{A.4}$$

where $\alpha, \beta \in (tA, tB, bA, bB)$ are layer and sublattice indices, and $\xi$ and $\xi'$ are the valley indices. Noticing that $H_{-1}$ is off-diagonal in the sublattice basis and diagonal in the layer and

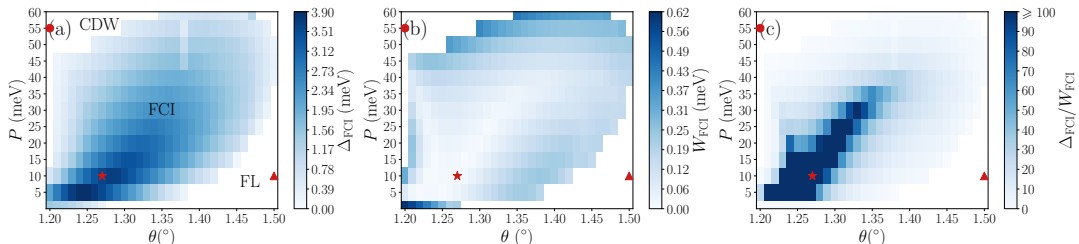

Figure 11: (a) The FCI gap $\Delta_{\text{FCI}}$, (b) FCI splitting $W_{\text{FCI}}$, and (c) their ratio $\Delta_{\text{FCI}}/W_{\text{FCI}}$ for $N = 10$ and $N_1 \times N_2 = 5 \times 6$ in valley $\xi = +$ with $w_1 = 110$ meV. The three-fold FCI degeneracy is absent in white regions. In (a), we give the tentative phase diagram in a single valley. The markers indicate the representative parameter points that we choose in Figs. 12 and 13.

valley basis, we can write down a typical term in $[H_{\text{int}}, H_{-1}]$ (up to some prefactor) as

$$\left[ \psi^\dagger_{\alpha,\xi}(\mathbf{k}_1)\psi^\dagger_{\beta,\xi'}(\mathbf{k}_2)\psi_{\beta,\xi'}(\mathbf{k}_3)\psi_{\alpha,\xi}(\mathbf{k}_4), \sum_{\mathbf{k}} \psi^\dagger_{(l,\sigma),\xi''}(\mathbf{k})\psi_{(l,\bar\sigma),\xi''}(\mathbf{k}) \right], \qquad (A.5)$$

where $\xi, \xi', \xi'' = \pm$ are the valley indices, $l = t, b$ is the layer index, and $\sigma = A, B$ is the sublattice index ($\bar\sigma = B, A$ if $\sigma = A, B$). We can then apply a general relation

$$\left[ c^\dagger_\alpha(\mathbf{k}_1)c^\dagger_\beta(\mathbf{k}_2)c_\gamma(\mathbf{k}_3)c_\delta(\mathbf{k}_4), \sum_{\mathbf{k}} f(\mathbf{k})c^\dagger_\mu(\mathbf{k})c_\nu(\mathbf{k}) \right]$$
$$= \delta_{\mu,\delta}f(\mathbf{k}_3)c^\dagger_\alpha(\mathbf{k}_1)c^\dagger_\beta(\mathbf{k}_2)c_\gamma(\mathbf{k}_3)c_\nu(\mathbf{k}_4) + \delta_{\mu,\gamma}f(\mathbf{k}_4)c^\dagger_\alpha(\mathbf{k}_1)c^\dagger_\beta(\mathbf{k}_2)c_\nu(\mathbf{k}_3)c_\delta(\mathbf{k}_4)$$
$$- \delta_{\nu,\alpha}f(\mathbf{k}_1)c^\dagger_\mu(\mathbf{k}_1)c^\dagger_\beta(\mathbf{k}_2)c_\gamma(\mathbf{k}_3)c_\delta(\mathbf{k}_4) - \delta_{\nu,\beta}f(\mathbf{k}_2)c^\dagger_\alpha(\mathbf{k}_1)c^\dagger_\mu(\mathbf{k}_2)c_\gamma(\mathbf{k}_3)c_\delta(\mathbf{k}_4), \quad (A.6)$$

to compute this term, where $c^\dagger$'s and $c$'s are general fermionic creation and annihilation operators, respectively, and $f(\mathbf{k})$ is an arbitrary function. The result is

$$\left[ \psi^\dagger_{\alpha,\xi}(\mathbf{k}_1)\psi^\dagger_{\beta,\xi'}(\mathbf{k}_2)\psi_{\beta,\xi'}(\mathbf{k}_3)\psi_{\alpha,\xi}(\mathbf{k}_4), \sum_{\mathbf{k}} \psi^\dagger_{(l,\sigma),\xi''}(\mathbf{k})\psi_{(l,\bar\sigma),\xi''}(\mathbf{k}) \right]$$
$$= \delta_{\xi,\xi''}\delta_{\alpha,(l,\sigma)}\psi^\dagger_{\alpha,\xi}(\mathbf{k}_1)\psi^\dagger_{\beta,\xi'}(\mathbf{k}_2)\psi_{\beta,\xi'}(\mathbf{k}_3)\psi_{(l,\bar\sigma),\xi''}(\mathbf{k}_4)$$
$$+ \delta_{\xi',\xi''}\delta_{\beta,(l,\sigma)}\psi^\dagger_{\alpha,\xi}(\mathbf{k}_1)\psi^\dagger_{\beta,\xi'}(\mathbf{k}_2)\psi_{(l,\bar\sigma),\xi''}(\mathbf{k}_3)\psi_{\alpha,\xi}(\mathbf{k}_4)$$
$$- \delta_{\xi,\xi''}\delta_{\alpha,(l,\bar\sigma)}\psi^\dagger_{(l,\sigma),\xi''}(\mathbf{k}_1)\psi^\dagger_{\beta,\xi'}(\mathbf{k}_2)\psi_{\beta,\xi'}(\mathbf{k}_3)\psi_{\alpha,\xi}(\mathbf{k}_4)$$
$$- \delta_{\xi',\xi''}\delta_{\beta,(l,\bar\sigma)}\psi^\dagger_{\alpha,\xi}(\mathbf{k}_1)\psi^\dagger_{(l,\sigma),\xi''}(\mathbf{k}_2)\psi_{\beta,\xi'}(\mathbf{k}_3)\psi_{\alpha,\xi}(\mathbf{k}_4)$$
$$= 0, \qquad (A.7)$$

leading to $[H_{\text{int}}, H_{-1}] = 0$. Therefore, only $H_{\text{kin}}$ contributes to $H_{\text{eff}}^{(2)}$ in our model, making $H_{\text{eff}}^{(2)}$ a non-interacting part in $H_{\text{eff}}$. This is different from the result in Ref. [49], in which new effective interaction terms appear in $H_{\text{eff}}^{(2)}$ for another Floquet lattice model. This difference is a result of different structures of the Hamiltonian between our model and the model in Ref. [49].

# B  Many-body physics with $w_1 = 110$ meV

In Sec. 4 of the main text, we have presented the results of the many-body physics in the $\nu = 1/3$ filled Floquet valence bands with $w_1 = 90$ meV. Now we repeat the investigations for

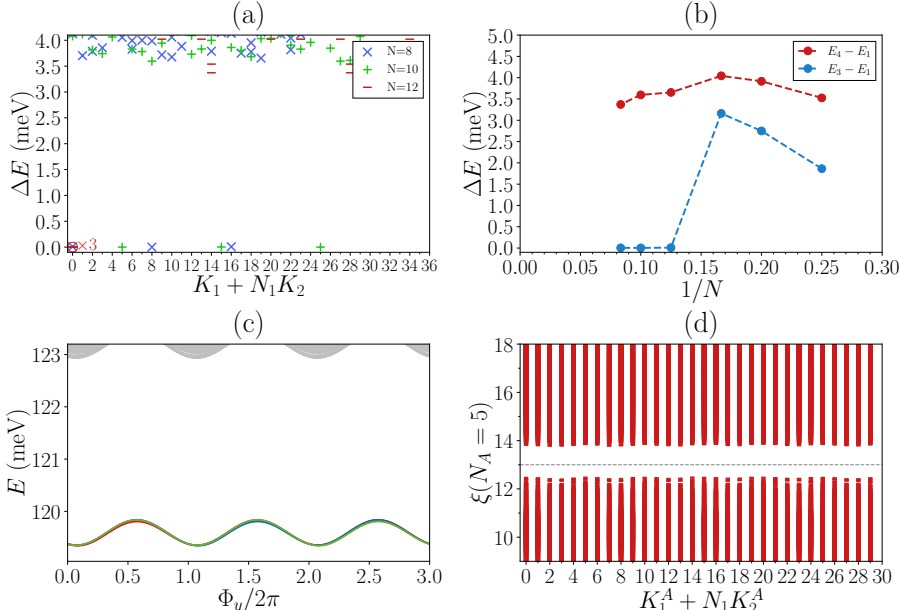

Figure 12: The $\nu = 1/3$ Laughlin FCIs in the Floquet valence band of valley $\xi = +$ at $\theta = 1.27°$, $P = 10$ meV when $w_1 = 110$ meV. (a) The low-lying energy spectra for $N = 8, 10, 12$ electrons on the $N/2 \times 6$ lattice. (b) The finite-size scaling of the energy gap $E_4 - E_1$ and the ground-state splitting $E_3 - E_1$ for $N = 4, 5, 6, 8, 10, 12$ electrons. (c) The spectral flow for $N = 10$, $N_1 \times N_2 = 5 \times 6$, where $\Phi_y$ is the magnetic flux insertion in the $\mathbf{a}_2$ direction. (d) The particle entanglement spectrum for $N = 10$, $N_1 \times N_2 = 5 \times 6$ and $N_A = 5$, with 23256 levels below the entanglement gap (the dashed line).

a stronger interlayer tunneling $w_1 = 110$ meV. Under the assumption of valley polarization, we find the overall phase diagram (Fig. 11) is very similar to that for $w_1 = 90$ meV. However, the FCI phase exists at larger twist angles, which are still small but obviously deviate from the magic angle $\sim 1.05°$. Similar to the case with $w_1 = 90$ meV, the stability of the FCI phase can be understood by studying the Berry curvature, the FS metric, and the bandwidth. The best three-fold degeneracies of ten electrons appear at $\theta \approx 1.20° - 1.35°$ and $P \approx 5$ meV $- 35$ meV [Fig. 11(c)]. We demonstrate the energy spectrum and the PES at a representative parameter point (labelled by the stars in Fig. 11) in this region (Fig. 12), where the energy gap protecting FCI ground states is even larger than that at $w_1 = 90$ meV. Meanwhile, the competition between FCI and CDW remains for $w_1 = 110$ meV. The signals of the CDW phase and the Fermi liquid phase, namely, the structure factor and the $\langle n(\mathbf{k}) \rangle - E(\mathbf{k})$ curve, are shown in Fig. 13 for two representative parameter points (labelled by the dots and triangles in Fig. 11, respectively).

Considering the similarity of the many-body phase diagram in a single valley between $w_1 = 90$ meV and $w_1 = 110$ meV, we expect the many-body physics in both cases is also similar when the assumption of valley polarization is relaxed, i.e., for $w_1 = 110$ meV a valley-polarized FCI phase exists and competes with a valley-polarized CDW phase and a $S_z = 0$ Fermi liquid phase.

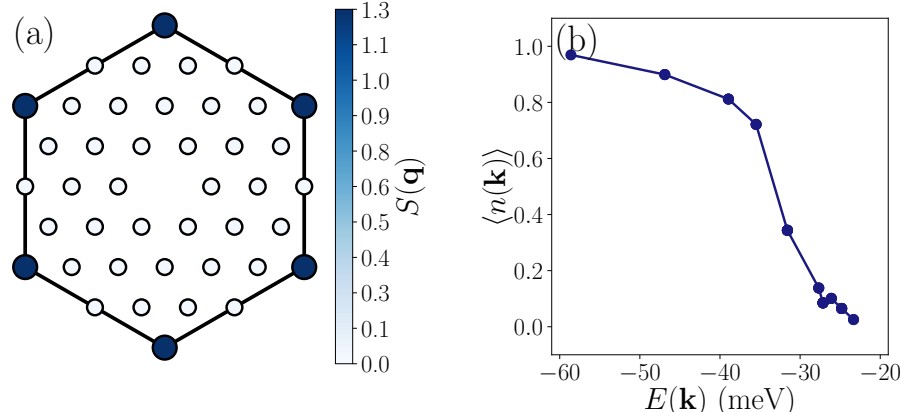

Figure 13: The competing phases of FCIs in the Floquet valence band of valley $\xi = +$ when $w_1 = 110$ meV. (a) Distribution of $S(\mathbf{q})$ in the MBZ for $N = 12$, $N_1 \times N_2 = 6 \times 6$ at $(\theta, P) = (1.20°, 55$ meV$)$. (b) The ground-state occupation $\langle n(\mathbf{k}) \rangle$ at momentum $\mathbf{k}$ as a function of band energy $E(\mathbf{k})$ for $N = 12$, $N_1 \times N_2 = 6 \times 6$ at $(\theta, P) = (1.50°, 10$ meV$)$.

## C Effects of time-reversal symmetry breaking on the many-body physics

The single-particle Hamiltonian of the static TBG aligned with hBN substrates on the top and at the bottom is

$$
\begin{aligned}
H'_{\text{kin}} = \sum_{\xi = \pm} \sum_{\mathbf{k}} & \left( \psi^\dagger_{t,\xi}(\mathbf{k}) h^\xi_{-\theta/2} \left( \mathbf{k} - \mathbf{K}^t_\xi \right) \psi_{t,\xi}(\mathbf{k}) + \psi^\dagger_{b,\xi}(\mathbf{k}) h^\xi_{\theta/2} \left( \mathbf{k} - \mathbf{K}^b_\xi \right) \psi_{b,\xi}(\mathbf{k}) \right) \\
& + \sum_{\xi = \pm} \sum_{\mathbf{k}} \sum_{j=0}^{2} \left( \psi^\dagger_{t,\xi} \left( \mathbf{k} - \xi \mathbf{q}_0 + \xi \mathbf{q}_j \right) T^\xi_j \psi_{b,\xi}(\mathbf{k}) + \text{H.c.} \right) \\
& + M \sum_{\xi = \pm} \sum_{\mathbf{k}} \left( \psi^\dagger_{t,\xi}(\mathbf{k}) \sigma_z \psi_{t,\xi}(\mathbf{k}) + \psi^\dagger_{b,\xi}(\mathbf{k}) \sigma_z \psi_{b,\xi}(\mathbf{k}) \right),
\end{aligned} \tag{C.1}
$$

where the term in the third row is a hBN-induced staggered potential of strength $M$ on the $A, B$ sublattices of each graphene layer. A rough estimation implies $M \approx 17$ meV [14], however, we will treat $M$ as a tunable parameter below. In this system, the alignment with hBN can lift the band touching at the Dirac points due to the breaking of the $\mathcal{C}_2$ symmetry [14]. However, the hBN alignment does not break the time-reversal symmetry, which means the Chern numbers of the isolated valence (or conduction) bands near the CNP in the two valleys must be opposite. Then the symmetry of the static TBG-hBN system is different from our driven TBG system in which the time-reversal symmetry is broken while the $\mathcal{C}_2$ symmetry is preserved, although in both cases the band gap can be opened at the Dirac points. We have studied the band gap, bandwidth, and the Chern number of the static TBG-hBN valence band in the $(\theta, M)$ parameter space. The results are very similar to those of the Floquet system in Figs. 2 and 3 once we replace $M$ with $P$ (note that in the topological region the static valence band carries $C = \mp 1$ in valley $\pm$). This is as expected, because in valley $\xi = + H'_{\text{kin}}$ only differs from the single-particle part of the effective Hamiltonian of the Floquet system by a tiny $H^{(2)}_{\text{eff}}$ term, and the band structure of the static TBG-hBN in valley $\xi = -$ is identical to that in valley $+$ upon a time-reversal conjugate.

We have seen that the static TBG-hBN and the driven TBG have different symmetries although the band gap is opened at the Dirac points in both systems. In order to study the

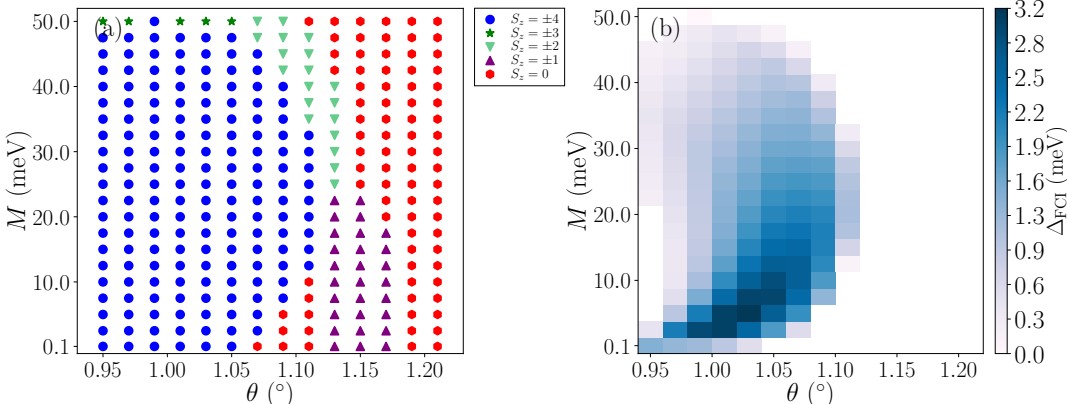

Figure 14: The many-body physics of $\nu = 1/3$ valence bands of static TBG-hBN when both valleys are considered for $N = 8, N_1 \times N_2 = 4 \times 6$. We choose $w_1 = 90$ meV. (a) The $S_z$ of the global ground state. (b) The generalized FCI gap.

effect of this symmetry difference on the many-body physics, we briefly explore the many-body physics in the static TBG-hBN system when the valence bands of the two valleys are partially occupied by electrons at total filling $\nu = 1/3$ and compare it to the driven TBG. Like what we did for the driven system, we project the screened Coulomb interaction to the two valence bands of the static TBG-hBN. We focus on the parameter region with nonzero band Chern number and choose $w_1 = 90$ meV. Under the assumption of valley polarization, we observe a robust Laughlin FCI phase in a single valley, competing with a CDW phase and a Fermi liquid phase. The phase diagram in a single valley is very similar to that of the driven TBG.

Then we relax the assumption of valley polarization. By contrast to the driven system, for static TBG-hBN we do not observe the Laughlin FCI phase in $S_z \neq \pm N/2$ sectors like those shown in Fig. 8. This is due to the opposite Chern number of the two valence bands in opposite valleys of static TBG-hBN. In Fig. 14(a), we display the $S_z$ quantum number of the global ground state of static TBG-hBN in the $(\theta, M)$ parameter space. It turns out that whether the time-reversal symmetry is broken does not significantly affect the valley polarization: Like in the driven system, there is a similar region in which the ground state of static TBG-hBN is valley polarized. The Laughlin FCI phase observed under the valley polarization assumption survives in this region. In Fig. 14(b), we show the generalized FCI gap defined in Sec. 4.2 for static TBG-hBN, namely the gap taking the intervalley excitations into account. For the system size of $N = 8$ electrons, the maximal FCI gap of static TBG-hBN is about two times larger than that in the driven case [Fig. 9(b)], however, a more precise comparison requires the access to larger system sizes. In Fig. 14(a), we also notice some regions with $S_z \neq \pm N/2$, which looks more complicated than Fig. 9(a). We will leave a detailed investigation of these competing phases to a future work.

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
