# Peer review of "Floquet Fractional Chern Insulators and Competing Phases in Twisted Bilayer Graphene"

_SciPost Physics, doi:SciPost Phys. 15, 148 (2023)_

## Round 2 · Referee Report · Anonymous (Referee 2) · 2023-8-7

Strengths

1- timely and relevant topic 2- state of the art numerics 3- appropriate justification of various hypothesis 4- interesting discussion of symmetries and comparison with similar models

Report

I thank the authors for their thorough answers to all my comments, and corresponding changes to the manuscript. I greatly appreciate the additions they made, which I believe have clearly improved the quality of the manuscript. This work now presents a very complete study of the emergence of FCIs in MATBG driven by circularly polarized light, and is fully worthy of publication.

---

## Round 2 · Author Response

Dear Editor,

We thank you for arranging the review of our manuscript, and for forwarding us the reports of the two Referees. We thank both Referees for their praise of our work and their valuable comments. We have carefully revised the manuscript to meet their concern. Below, we reply to the Referees’ comments point by point. Thank you very much again for your time and consideration.

Sincerely, Peng-Sheng Hu, Yi-Han Zhou, and Zhao Liu

Response to the First Referee

Referee's comment 1: "The authors study magic angle twisted bilayer graphene (TBG) subject to a drive of circularly polarized light as a platform for the realization of fractional Chern insulators (FCIs), i.e. the fractional quantum Hall effect in the absence of a static magnetic field. They use Floquet theory to obtain an effective static Hamiltonian under the drive of frequency Ω. Up to second order in 1/Ω, they find that the drive does not affect the interaction term of the Hamiltonian (screened Coulomb interaction), while its main effect on the kinetic part of the Hamiltonian is the addition of a staggered sublattice potential of strength P akin to the (static) effect of alignment of graphene to the hexagonal boron nitride (hBN) substrate. A similar observation had been reported in previous Floquet treatment of TBG up to first order; the current manuscript clarifies that the second order term does significantly affect the Floquet Hamiltonian (no additional the interaction term, small - 4% - kinetic term). After deriving the Floquet Hamiltonian, the authors numerically study its many-body phase diagram at filling fraction ν = 1/3, as a function of the twist angle theta, and staggered potential P . They assume full valley and spin polarization, and project the interaction to a single band. The nature of the many-body ground state is carefully analyzed base on particle entanglement spectroscopy, flux insertion, static structure factor, and occupation of single-particle states, with appropriate finite-size extrapolation. The authors find three different phases: a FCI, a charge density wave (CDW), and a Fermi liquid. These different phases are the same ones already identified in other FCI models, especially TBG models in the absence of a drive.

I find the motivation for the present study to be relevant and timely, and the paper is clearly written, with convincing numerical evidence. Yet, some key elements are missing, in my opinion, in order to fully answer the questions the authors set out to address. If these elements were provided satisfyingly, I would recommend publication."

Response 1: We thank the Referee for his/her praise of our manuscript and the valuable comments which we think are very helpful for improving our manuscript. We address these comments below.

Referee's comment 2: "First, unlike hBN alignment, the circularly polarized drive explicitly breaks time-reversal symmetry (TRS). Therefore, I would expect some differences between the band structures of TBG + staggered AB-potential and TBG + staggered AB- potential + TRS breaking. Can the authors please clarify? This question has likely been explored in previous references on the floquet theory of TBG, but it would be relevant to make it explicit in the context of the many-body phase diagram. This is especially important because it could potentially affect valley polarization (assumed in the paper), which emerges spontaneously from strong interactions at some fillings in the TRS case."

Response 2: We appreciate the Referee for raising this important issue. The Dirac cones of static TBG is protected by the C2*T symmetry, where C2 is the 180-degree in-plane rotation, and T is the time reversal. Either alignment with hBN or light driving can open the band gap at the Dirac points, however, the mechanisms of symmetry breaking are different. The hBN alignment breaks the C2 symmetry but preserves the T symmetry. The light driving breaks the T symmetry but preserves the C2 symmetry.

For the driving frequency and driving strength considered in our work, the single-particle Hamiltonian of the static TBG-hBN only differs from the effective single-particle Hamiltonian of the driven TBG by a very small second-order term H_{eff}^(2) in valley ξ = +. Therefore, their band structures, including the band gap, bandwidth, and band Chern numbers are very similar. In the other valley ξ = −, the band energies can always be related to those in valley ξ = + under a k → −k transformation, which is a result of the remaining T (for static TBG-hBN) or C2 (for driven TBG) symmetry. Hence the band energies of static TBG-hBN and driven TBG in valley ξ = − are again very similar. However, for driven TBG, we expect the Chern numbers of the valence (or conduction) bands in the two valleys are the same due to the breaking of T symmetry. By contrast, for static TBG-hBN these Chern numbers should be opposite in the two valleys because the T symmetry is preserved. Therefore, when the band gap is opened at the Dirac points, the Chern number of the valence (or conduction) band of driven TBG are opposite to that of static TBG-hBN in valley ξ = −.

In summary, the band energies of static TBG-hBN and driven TBG are very similar in both valleys for the driving frequency and driving strength considered in our work. However, when the band gap is opened at the Dirac points, their valence band (or conduction band) Chern numbers have a sign difference in valley ξ = −, which is a result of T symmetry breaking in the driven case. We have numerically confirmed this conclusion. In the revised manuscript, we have included both TBG valleys to analyze the symmetry of the single-particle Hamiltonian in Sec. 2.

To probe the effect of T symmetry breaking in the many-body level, we relax the assumption of valley polarization to study the many-body physics when the valence bands in two valleys are partially occupied at total filling ν = 1/3. In this case, the valley pseudospin Sz = (N+ − N−)/2 is a good quantum number, where N± is the number of electrons in valley ξ = ±. For driven TBG, we observe the Laughlin FCI phase in each Sz sector. They originate from the Laughlin ferromagnetism in the chiral limit of w0 = 0, where w0 is the interlayer tunneling within the same sublattice. The FCI states in different Sz sectors should be related by pseudospin ladder operators. On the contrary, for the static TBG-hBN, the stable Laughlin FCI phase only exists in the valley polarized sectors. This is because the two valleys carry opposite Chern numbers, so the Laughlin state is destroyed once both valleys are occupied by electrons. However, for both systems, we find a similar region in the phase diagram where the valley-polarized Laughlin FCI is the global ground state. So it turns out that whether the time-reversal symmetry is broken does not affect the valley polarization of the ground state too much. This observation also justifies the assumption of valley polarization in a large region of the phase diagram. We have added new sections: Sec. 4.2 and Appendix C in the revised manuscript to include these results.

Referee's comment 3: "Second, the authors have provided only small hints to microscopically explain the many-body phase diagram. As the authors are well aware, there are a few single-particle markers that can predict the emergence of a FCI in a Chern band: small enough deviation from a homogeneous Berry curvature, small enough deviation from the saturation of the trace inequality of the quantum metric tensor, and small enough bandwidth (the bandwidth is provided, but not the others). I think it would be helpful to compare the maps of these markers with the obtained phase diagram, to provide a more systematic understanding of their many-body results."

Response 3: We thank the Referee for pointing this flaw out. We have computed the Berry curvature fluctuation σ(B) and the deviation δ_{tr} from the saturation of the trace inequality of the quantum metric tensor in the parameter space. The results are shown in Fig. 7 of the revised manuscript, with some discussions at the end of Sec. 4.1. Indeed, we find the strongest FCI phase appears when both σ(B) and δ_{tr}, as well as the bandwidth, are sufficiently small. Nevertheless, the minima of these three quantities are not located at the same parameter point, so over-minimizing an individual quantity does not necessarily improve the stability of the FCI phase.

Referee's comment 4: "Finally, my last comments concern the FCI many-body gap. The authors have provided a value of around 20 Kelvins. What Coulomb interaction strength U was used to obtain this number? The experimental U may not be known to a great precision, so it may be more useful to express the energy spectrum and many-body gap in units of U. I am also surprised by this value, which is ten times larger than the value obtained in other references of FCIs in TBG (with no drive), using the value U ∼ 20 meV. The discrepancy may be explained by the U value used by the authors, if not it would be important to understand its origin."

Response 4: We thank the Referee for this comment. We can estimate the Coulomb interaction strength as U=e^2/(4πε a_M), where a_M = a/(2sin(θ/2)) with a the lattice constant of monolayer graphene and we choose ε = 4ε0 with ε0 the dielectric constant of vacuum. Under the valley-polarization assumption, we have U ≈ 26.8 meV at the representative parameter point (θ, P ) = (1.05◦, 10 meV) in the stable FCI phase of our driven TBG system (see Fig. 5 in the revised manuscript). The FCI many-body gap extrapolated to the thermodynamic limit at this parameter point is about 2 meV ≈ 0.07U. Because we include the band dispersion in the many-body simulation, the many-body gap is not simply proportional to the interaction strength. Once we relax the valley polarization assumption, this gap could be reduced by a factor of 2 (see Fig. 9b in the revised manuscript). Note that this energy gap in general changes with the filling factor and the choice of active bands (valence versus conduction bands). It can be further reduced by spinful excitations and disorder.

We have also compared our results to other references of FCIs in static TBG, for example, Ref. [20] (Chern bands of twisted bilayer graphene: Fractional Chern insulators and spin phase transition, by Cécile Repellin and T. Senthil, https://doi.org/10.1103/PhysRevResearch.2.023238). In that paper, the authors assumed valley polarization but relaxed spin polarization. Another difference from our work is that they considered the conduction band at electron filling ν = 1/3. They found the FCI gap in the spin-polarized sector is about 0.01U. We notice that their definition of U is probably larger than us by a factor of 2π, because the Fourier transform of the bare Coulomb potential was written as U/q instead of U(2π/q) in their paper. If this is true, the FCI gap in the ν = 1/3 conduction band is in the same order of magnitude as our result in the valence band.

In the Sec. 4.1 of the revised manuscript, we have clarified the Coulomb interaction strength U used in our paper and expressed the typical FCI gap in terms of U .

Referee's comment 5: "Additionally, do Floquet FCIs require larger gaps (compared to intrinsic FCIs) in order to be observed experimentally (due to heating processes for example)? The authors state that the estimated gap is an order of magnitude larger than the ones that can be observed experimentally, so it would be good to clarify if this typical estimate of what is detectable takes into account the non-equilibrium nature of the system."

Response 5: We thank the Referee for this comment. Indeed heating is a main challenge for realizing Floquet FCIs. We are not sure whether Floquet FCIs require larger gaps to overcome the heating processes. A general strategy to overcome heating is to increase driving frequency, so that the life of the prethermal regime exponentially grows with the frequency. In this prethermal regime, the behavior of the system is well captured by the effective Hamiltonian obtained from the Magnus expansion. We have added a statement at the end of the discussion in the revised manuscript.

Response to the second Referee

Referee's comment 1: "In the light of the points mentioned unter Strenghts/Weaknesses, I consider this a nice and timely piece of work that deserves publication in SciPost. Nevertheless, there are two points that need clarification in my view:"

Response 1: We thank the Referee for his/her praise of our manuscript. Meanwhile, he/she raised two issues, to which we respond below.

Referee's comment 2: "How is the assumption of valley and spin polarization justified? The authors assume this from the outset but do not comment on this."

Response 2: We appreciate the Referee for raising this important issue. Indeed it would be nice to include both valley and spin degrees of freedom to explore the many-body physics. However, that will greatly increase the Hilbert space dimension, limiting the exact diagonalization calculations to very small system sizes. In order to reach reasonable system sizes, we have to assume that some degrees of freedom are frozen. For static TBG, previous studies found that interactions are able to polarize spin under suitable circumstances (see Refs. [10, 59–62]). For driven TBG, the two-fold spin degeneracy of each band is the same as in static TBG, so we expect that at least spin polarization also exists for driven TBG under suitable circumstances. In fact, interactions can also induce valley polarization in static TBG. However, the time-reversal symmetry between two valleys is broken when the system is driven by light. Therefore, light driving could lead to different physics in the two valleys from the static TBG. We add a paragraph above Eq. (3) in Sec. 2.1 to clarify this.

In the revised manuscript, we keep the valley degree of freedom when deriving the effective Floquet Hamiltonian, and examine whether the valley polarization holds for driven TBG. We explore the many-body physics when both valleys are included. Remarkably, we find a large valley-polarized region in the phase diagram of driven TBG, and the CDW phase and the strongest FCI phase which are observed under the valley polarization assumption are included in this region. In this sense, our assumption of valley polarization is reasonable. We also identify an intervalley Fermi liquid phase, where electrons equally occupies the two valleys. The results are presented in the new Sec. 4.2.

Referee's comment 3: "If I understand correctly, Fig. 3 is not the result of an interacting calculation. Why is it then not presented as a nice phase diagram like Fig. 2, but only so few points are calculated?"

Response 3: We thank the Referee for this question. Indeed, Fig. 3 is the Chern number of the non-interacting valence band in valley +. Since the Chern number cannot change as long as the band gap does not close, previously we only drew some points to highlight the change of Chern number when the band gap vanishes. In the revised manuscript, we have added more points in Fig. 3 to show the Chern number in the entire parameter space.

---

## Round 2 · List of Changes

• In the paragraph above Eq. (3), we add an argument for the assumption of spin polarization.
• We now include both valleys to derive the effective Hamiltonian of our driven system in Sec. 2 and Appendix A.
• In the two paragraphs below Eq. (9), we clarify the symmetries of our driven model, especially the time-reversal symmetry breaking.
• We update Fig. 3 to present denser data points of the Chern number.
• In the third paragraph of Sec. 4.1, we give an estimation of the Coulomb interaction strength U in our system, and express the FCI gap at a representative parameter point in terms of U.
• We add Fig. 7 and the last paragraph of Sec. 4.1 to show results of the quantum geometry of the Floquet valence band.
• We add a new section 4.2. In this section, we relax the assumption of valley polarization to explore the many-body physics when electrons are allowed to occupy both valleys, and justify the assumption of valley polarization in a large region of the phase diagram.
• At the end of Sec. 5, we add some discussions on the issue of heating process.
• We add Appendix C to briefly explore the many-body physics in static TBG-hBN and compare with the driven case.
• Other small changes to improve the presentation.

---

## Editorial Decision

published